# A Landsat derived annual inland water clarity dataset of China between 1984 and 2018

Hui Tao[1, 2], Kaishan Song[1, 3*], Ge Liu[1], Qiang Wang[1], Zhidan Wen[1], Pierre-Andre Jacinthe[4], Xiaofeng Xu[1], Jia Du[1], Yingxin Shang[1], Sijia Li[1], Zongming Wang[1], Lili Lyu[1], Junbin Hou[1], Xiang Wang[1], Dong Liu[5], Kun Shi[5], Baohua Zhang[3], Hongtao Duan[5*]

[1]Northeast Institute of Geography and Agroecology, Chinese Academy of Science, Changchun, 130102, China
[2]University of Chinese Academy of Sciences, Beijing, 100049, China
[3]College of Urban Research and Planning, Liaocheng University, Shandong, China
[4]Department of Earth Sciences, Indiana University Purdue University, Indianapolis, IN, USA
[5]Nanjing Institute of Geography and Limnology, Chinese Academy of Science, Nanjing 210008, China

*Correspondence to*: Kaishan Song (songkaishan@iga.ac.cn); Hongtao Duan (htduan@niglas.ac.cn)

**Abstract.** Water clarity serves as a sensitive tool for understanding the spatial pattern and historical trend in lakes trophic status. Despite the wide availability of remotely-sensed data, this metric has not been fully explored for long-term environmental monitoring. To this end, we utilized Landsat top of atmosphere reflectance products within the Google Earth Engine in the period 1984-2018 to retrieve the average SDD for each lake in each year. Three Secchi disk depth (SDD) datasets were used for model calibration and validation from different field campaigns mainly conducted during 2004-2018. The red/blue band ratio algorithm was applied to map SDD for lakes (> 0.01 km³) based on the first SDD dataset, where $R^2$ = 0.79, rRMSE = 61.9%. The other two datasets were used to validate the temporal transferability of the SDD estimation model, which confirmed the stable performance of the model. The spatiotemporal dynamics of SDD were analysed at the five lake regions and individual lake scales, and the average, changing trend, lake number and area, and spatial distribution of lake SDDs across China were presented. In 2018, we found the number of lakes with SDD < 2 m accounted for the largest proportion (80.93%) of the total lakes, but the total area of lakes with SDD of <0.5 m and > 4 m were the largest, both accounting for about 24.00% of the total lakes, respectively. During 1984-2018, lakes in the Tibetan-Qinghai Plateau region (TQR) had the clearest water with an average value of 3.32±0.38 m, while that in the Northeastern region (NLR) exhibited the lowest SDD (mean: 0.60±0.09 m). Among the 10,814 lakes with SDD results more than 10 years, 55.42% and 3.49% of lakes experienced significant increasing and decreasing trends, respectively. At the five lake regions, except for the Inner Mongolia-Xinjiang region (MXR), more than half of the total lakes in every other region exhibited significant increasing trends. In the Eastern region (ELR), NLR and Yungui Plateau region (YGR), almost more than 50% of the lakes that displayed increase or decrease in SDD were mainly distributed in the area range of 0.01-1 km², whereas that in the TQR and MXR were primarily concentrated in large lakes (> 10 km²). Spatially, lakes located in the plateau regions generally exhibited higher SDD than those situated in the flat plain regions. The dataset can now be accessed through the website of the National Tibetan Plateau Data Center (http://data.tpdc.ac.cn): DOI: 10.11888/Hydro.tpdc.271571.

## 1 Introduction

Lakes and reservoirs are important aquatic habitats and serve as freshwater water sources for drinking, industrial and agricultural uses (Pekel et al., 2016; Tranvik et al., 2009; Wetzel, 2001). More than 26,000 lakes (with area > 0.01 km²) and 78,000 reservoirs are distributed across China (Song et al., 2018), providing multiple ecosystem services (Feng et al., 2019b; Lehner and Doll, 2004; Tranvik et al., 2009; Yang and Lu, 2014). Over the last four decades, China has made considerable achievements with respect to socio-economic development but has also faced increasing water pollution challenges due to, among other contributing factors, agricultural nonpoint pollution, wastewater discharge, urban expansion, and increased water consumption (Han et al., 2016; Qin et al., 2010; Tong et al., 2017). Eutrophication and algal blooms proliferation are the clearest manifestations of these water quality problems, and major efforts have been made (afforestation, conversion of cropland to grassland or wetland) to mitigate these impacts and restore the ecological integrity of inland water systems (Huang et al., 2016; Ma et al., 2020; Tong et al., 2020).

Across the country, the number of stations dedicated to the monitoring of water quality in lakes (59) and reservoirs (52) is very limited in comparison to the national inventory of lakes and reservoirs (SOEE, 2018). Water resource managers in China clearly need better assessment tools to monitor inland water quality (Rosenzweig et al., 2011). Commonly expressed as Secchi disk depth (SDD) (Carlson, 1977), water clarity provides both a practical and comprehensive measure of the trophic state of aquatic ecosystems (Olmanson et al., 2008; Richardson et al., 2010). However, traditional SDD measurements are limited in terms of their suitability for monitoring large water bodies exhibiting strong spatiotemporal dynamics (Kloiber et al., 2002; Song et al., 2020). Although a Secchi disk apparatus is easy to operate in the field, water clarity monitoring in lakes or reservoirs (herein lakes) located in remote areas could be nearly impossible without aquatic vehicles and may not yield data with sufficient spatial and temporal frequency necessary for trend analysis (Kloiber et al., 2002; Olmanson et al., 2008).

The abundance of optically-active constituents (OACs; phytoplankton, non-algal particles and CDOM) is related to the trophic status of aquatic ecosystems, and also contributes to water clarity and water surface reflectance which can be captured by space-borne sensors (Gordon et al., 1983; Lee et al., 2015). Remote sensing has been widely used for monitoring the spatiotemporal dynamics of SDD at regional and national scales. Available methods for SDD estimation using remote sensing data can be grouped into three categories: analytical, semi-analytical, and empirical algorithms (Doron et al., 2007; Lee et al., 2015; Liu et al., 2020b; McCullough et al., 2013; Olmanson et al., 2008; Olmanson et al., 2011). The first two methods are difficult to apply to large-scale studies (provincial and national scales) due to the complex theoretical models and parameterization processes, and expensive equipment required (Cao et al., 2017; Giardino et al., 2007). The last group of methods is widely used to retrieve SDD at multiple scales due to its simplicity and operability (Duan et al., 2009; Feng et al., 2019a; McCullough et al., 2012; Olmanson et al., 2011; Shen et al., 2020).

In the past, we faced the challenge of how to handle and analyse big data at national or global scales, like remote sensing datasets from different satellites. Since 2010, Google has launched the big geo data platform based on cloud computing,

named Google Earth Engine (GEE), which is time-saving for users to do some scientific researches online (such as vegetation, agriculture, hydrology, land cover and other applications) without downloading these satellite images (Amani et al., 2020). The GEE platform mainly comprises datasets of remote sensing, geophysics and meteorology. The remote sensing datasets contain Landsat (1972-present), Moderate Resolution Imaging Spectrometer (MODIS; 2000-present) and Sentinel (2014-present) (https://code.earthengine.google.com/). Remote sensing images are selectively used to estimate SDD for specific regions according to their spatial and temporal resolutions, among which the Landsat images can not only be used to examine the long-term (3-4 decades) spatiotemporal variation of SDD but also monitor lakes ranging from the small to the large with its higher spatial resolution (30 m). Therefore, the GEE platform is an optimal choice to quickly map SDD long time-series dynamics based on Landsat observation across China.

In recent years, a few studies have examined the spatiotemporal dynamics of SDD in lakes across China, but they mainly focused on the large lakes and reservoirs (area >10 $km^2$) (Liu et al. 2020a; Wang et al. 2020a; Zhang et al. 2021). Smaller lakes (area < 10 $km^2$) are widely distributed across the country, but our understanding of their ecological status remains limited. For example, Liu et al. (2020a) used an empirical model and the MODIS red and green bands (2000-2018) within GEE to study SDD variation in 412 large lakes (area > 20 $km^2$) across China. Wang et al. (2020a) applied water color parameters (Forel-Ule Index and hue angle) to MODIS data (2000-2017) and obtained SDD data for 153 lakes (> 25 $km^2$) across China. Zhang et al. (2021) built a simple power function model based on Landsat red band(2016−2018)to investigate the spatial distribution of SDD in 641 lakes (≥10 $km^2$) across China. In addition, other investigations of the spatio-temporal variations of SDD have been made using MODIS data for lakes in the Yangtze Plains (50 lakes, >10 $km^2$; Feng et al., 2019a) and in the Tibetan Plateau (64 lakes, >50 $km^2$; Pi et al., 2020). In these studies, the empirical models exhibited better ability than other models to estimate SDD at large-scales.

In this study, we tuned a recently-developed SDD empirical model which has been demonstrated as effective to map the spatial-temporal dynamics of SDD in surface waters based on atmospherically-corrected Landsat reflectance products in GEE (Song et al., 2020). The overall purpose of this study was to map the spatiotemporal variation of SDD in lakes (> 0.01 $km^2$) across China from 1984 to 2018. Specifically, the objectives were to: (1) built a lake SDD estimation model across China based on extensive *in-situ* measurements; (2) derive SDD of lakes across China using Landsat data embedded in GEE; (3) analyse the inter-annual variability of SDD at the lake regions scale and the individual lake scale. Such a research would provide valuable information regarding water quality conditions and inform future water resources planning and management.

## 2 Study area

China is a vast and physiographically-diverse country endowed with a large number of lakes. Based on broad regional variations of landforms and climate characteristics, the lakes in China have been grouped into five regions (Ma et al., 2011) (Fig. 1a). The Inner Mongolia-Xinjiang lake region (MXR) and Tibetan-Qinghai Plateau lake region (TQR) are located in

arid or semiarid climates, while the Northeastern lake region (NLR), Yungui Plateau lake region (YGR) and Eastern lake
       region (ELR) are situated in the Asian monsoon climate zone. The MXR and TQR regions have lower annual precipitation,
       lower temperature and higher evaporation level than other three lake regions. Regionally, lakes distribution sourced from
       Song et al. (2020) is as follows (in decreasing order): 49% in ELR, 22% in NLR, 18% in YGR, 8% in MXR and 4% in TQR
       (Fig. 1b). However, on the basis of lake surface area, regional distribution is slightly different and is in the order: TQR
(41%) > ELR (30%) > MXR (14%) > NLR (10%) > ELR (6%) (Fig. 1b). The lakes in the plateau region with higher
       elevation are less affected by human activities, and generally exhibit better ecological condition than lakes in the other
       regions (Zhang et al., 2019). In contrast, the lakes in the plain regions are frequently influenced by anthropogenic activities,
       such as urbanization, population growth, agricultural fertilizer and wastewater discharge (Feng et al., 2019a; Tong et al.,
       2020).

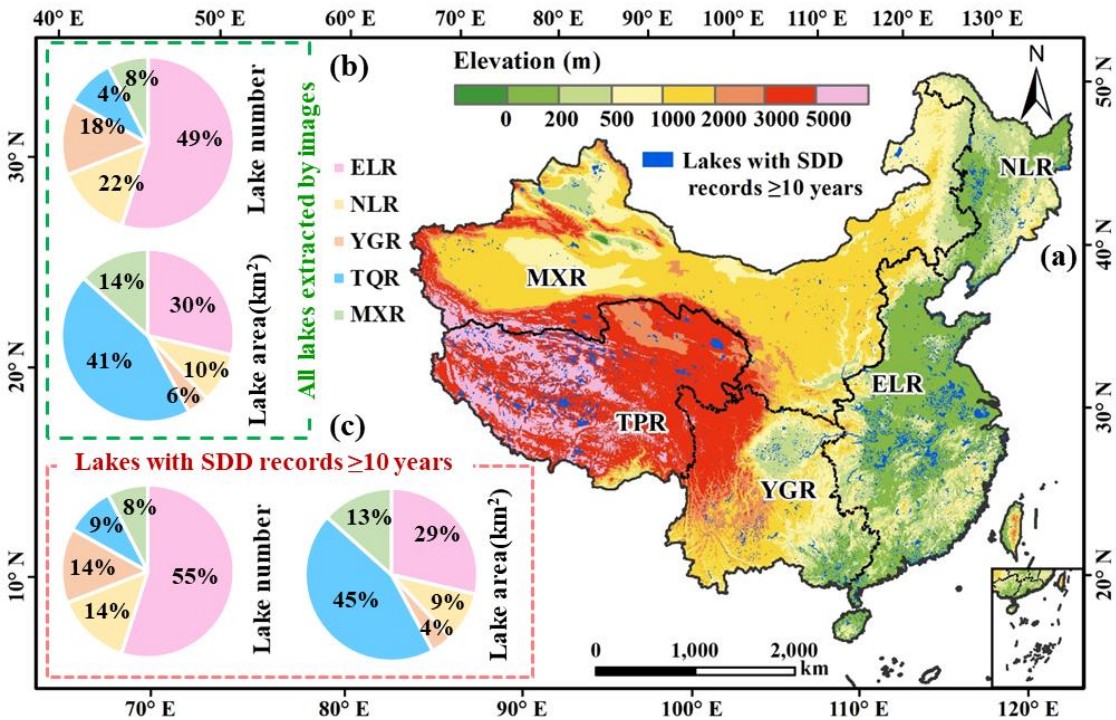

**Figure 1: The geographical distribution of lakes with water clarity (SDD) records of more than 10 years (lake area > 0.01 km²; N=11336) in different lake regions across China (a). The percentage distribution of lakes, based on the number of lakes and lakes surface area in the five lake regions is shown in the pie charts. The left one (green box) shows about all lakes extracted from Landsat images (b), while the lower left corner one (red box) displays about lakes with SDD records more than 10 years (c).**

## 3 Methods

### 3.1 Waterbody mask

Following Song et al. (2020), the lake boundaries (lakes and reservoirs) with area > 0.01 km$^2$ across China were derived from Landsat 8 OLI images mainly acquired in 2016, and detailed description on boundary extraction is available in that study. However, some lakes in China have changed substantially over time. These lakes were dealt with separately to obtain their boundaries in each year during the period 1984-2018. To obtain the information of lake area variation (e.g., size and year), we referred to an analysis on multi-decadal lake area (≥1 km$^2$ in size) changes in China during 1960s–2015 (Zhang et al., 2019) (Fig. S1). The datasets of lake boundaries (1960s-2020) have been released by the National Tibetan Plateau Data Centre. As for the reservoirs, we mainly viewed and compared the Landsat natural color images on the website of Earthdata Search (https://search.earthdata.nasa.gov/) and historical images embedded in Google Earth to confirm the changing region, respectively. For the small lakes with area < 1 km$^2$ obtained from the study of Song et al. (2020) we assumed their boundaries to remain unchanged during the study period.

We extracted the boundaries of these changing lakes using Landsat images during 1984-2018. The cloudless TOA image of each path and row was downloaded from the GEE platform, and processed to obtain the Modified Normalized Difference Water Index (MNDWI) as follows:

$$\text{MNDWI} = (R_{rc,Green} - R_{rc,SWIR})/(R_{rc,Green} - R_{rc,SWIR}) \tag{1}$$

where $R_{rc,Green}$ and $R_{rc,SWIR}$ are the Rayleigh scattering reflectance in the green band and short-wave infrared (SWIR) band, respectively. First, we used the MNDWI, combined with Tasseled Cap Transformation (TC) and a density slicing with multi-threshold approach, to build a decision tree for extracting water body boundaries using the ENVI software package (Rokni et al., 2014; Xu et al., 2006). Then, Landsat images acquired during 1984-2018 were classified into water and non-water areas (Feyisa et al., 2014; Wang et al., 2020b). The extracted water bodies were subsequently converted into polygons with contiguous pixels and stored in shape file format using the ArcGIS10.4 (ESRI Inc. Redlands, CA, USA). According to the shoreline features, we divided water bodies into lakes, reservoirs and rivers. By referring to the Global Reservoirs and Dams database (Lehner et al., 2011), Chinese Reservoirs and Dams database and high-resolution images from Google Earth, we distinguished rivers and reservoirs from water bodies mainly by visual interpretation. The shape file of lakes and reservoirs (herein lakes) was used as a water mask to extract the SDD map derived from the Landsat imageries (Fig. 1a).

The impact of land contamination on water remains a challenge in accurately retrieving water quality parameters (Jensen. 2006; Hou et al., 2018; Liu et al., 2020a; Wang et al., 2020a). Jensen. (2006) pointed out that the different surface objects have different reflectance to NIR band. For instance, land and vegetation can largely reflect the NIR band strongly absorbed by water, especially for the shallow lakes or reservoirs. In our study, one pixel (two pixels) buffer inward of water boundary was removed for lakes with area ≤ 1 km$^2$ (> 1 km$^2$) in order to avoid the influence of adjacent land on water bodies that can result in mixed land-water pixels. The determination of the number of pixel buffered was referenced to the method proposed in the study of Wang et al. (2018) who made a comparison of water-leaving reflectance in the transects selected from the

land-water boundaries to identify a suitable buffer distance. This method has been demonstrated to be effective in other studies related to SDD estimation (Liu et al., 2020a; Wang et al., 2020a).

## 3.2 SDD in-situ data collection across China

We used three SDD datasets for model calibration and validation (Fig.2a). To assemble the first dataset (IGA-04-19), we conducted 37 field campaigns from April 2004 to September 2018, surveyed 361 water bodies and collected 2,293 samples from lakes and reservoirs across China (Table S1), most of which were collected in late summer and early autumn. The second dataset was assembled from field campaigns (2007-2009) conducted by researchers from the Nanjing Institute of Geography and Limnology, Chinese Academy of Sciences. The third dataset (229 samples) was collected by different research groups during 1980s-1990s, and included records for which data collection date was not available. The spatial distribution of these three groups samples is shown in Fig. 2a. At each station, Secchi disk depth (SDD, in cm) was determined to represent water clarity, and was taken as the depth from water surface where a black-white Secchi disk can no longer be seen under water. For the first two datasets, SDD data derived from field surveys (2004-2018) were matched with the top of atmosphere reflectance (TOA) data collected by Landsat satellites overpassing a lake/reservoir within 7 days of field site visit, and the average reflectance of pixels within a $3 \times 3$ window matching a sampling point was extracted for bands 1-5 (Kloiber et al., 2002). After matching the *in-situ* SDD with images, altogether, 1,301 and 340 pairs of data were obtained based on the first and second SDD datasets, respectively. For the third dataset, the cloud-free TOA images whose dates were closest to time recorded on the lake survey reports were selected to match the measured SDD, which were between May and October during the period of field survey. Finally, 229 match-ups were found by expanding the time window between the third dataset of SDD and images.

## 3.3 Acquisition and processing of Landsat imagery data

To track the dynamics of lakes SDD in the past 35 years, all available Landsat TM/ETM+/OLI images of TOA across China were used in this study (~82,000 images, >60 terabytes of data) via GEE platform. The number of images used for SDD estimate in a specific year spanned a large range, from 371 in 1984 to 4,784 in 2018 (Fig. 2b), with more images available when two satellites operated simultaneously in space to acquire Landsat imagery. In this study, based on the GEE platform, the TOA images were mainly collected during the ice-free season (May to October) from 1984 to 2018 in the TQR, MXR, NLR, and ELR, except for YGR (from January to December) due to lack of good-quality images. The pixel_qa band, as a pixel quality control band generated from the CFMASK algorithm, was selected to mask out the land and snow/ice, and to remove cloud contamination (cloud cover >60%) in the GEE platform, thus minimizing the potential impact of cloud on SDD estimation accuracy. Landsat imagery atmospheric correction is a key step for water quality inversion (Wang et al., 2009), particularly for monitoring of temporal variation at large scale. The TOA products within GEE were produced using the equations developed by Chander et al. (2009), and the function of these equations was to convert calibrated digital numbers to absolute units of TOA reflectance. The description of Landsat TOA products is available on the GEE platform

(https://developers.google.com/earth-engine/datasets/catalog/landsat). More than 98.35 % of the pixels within China had a total of qualified observations > 35 in the past 35 years, and the majority of images had more than three scenes of good observations for each year.

### 3.4 Model for SDD estimation and mapping in GEE

Model development was a key step in this study. For the first match-ups dataset, i.e., 1,301 pairs of *in-situ* SDD and TOA,
we divided the valid data into four groups, with three groups used to calibrate the model (N= 976) and one group (N = 325) used for model validation. Based on a previous investigation, the red and blue (or green) band ratio was found to improve the performance of reflectance-based water quality models both in terms of their spatial and temporal transferability (Kloiber et al., 2002; Olmanson et al., 2008; Song et al., 2020). Thus, by trying the band combination, the red/blue band ratio algorithm using the first matched dataset was employed in this study to map SDD of water bodies, and was mathematically expressed
as:

$$\text{Ln (SDD)} = -5.6828 \times (\text{Red/Blue}) + 7.8413, \tag{2}$$

Then, combining the aforementioned image-processing methods, Eq. (1) was applied to the TOA images from 1984 to 2018 to estimate the SDD in the lakes with an area > 0.01 km² over China via the GEE platform. The annual mean SDDs at the pixel scale were obtained by averaging all available estimated results, and then the lake-based annual mean SDDs had been
further worked out. During the calculations, we only took into consideration lakes with SDD results of more than 10 years. At last, 10,814 lakes (size > 0.01 km²) were examined for the interannual dynamics of SDD (Fig.1c).

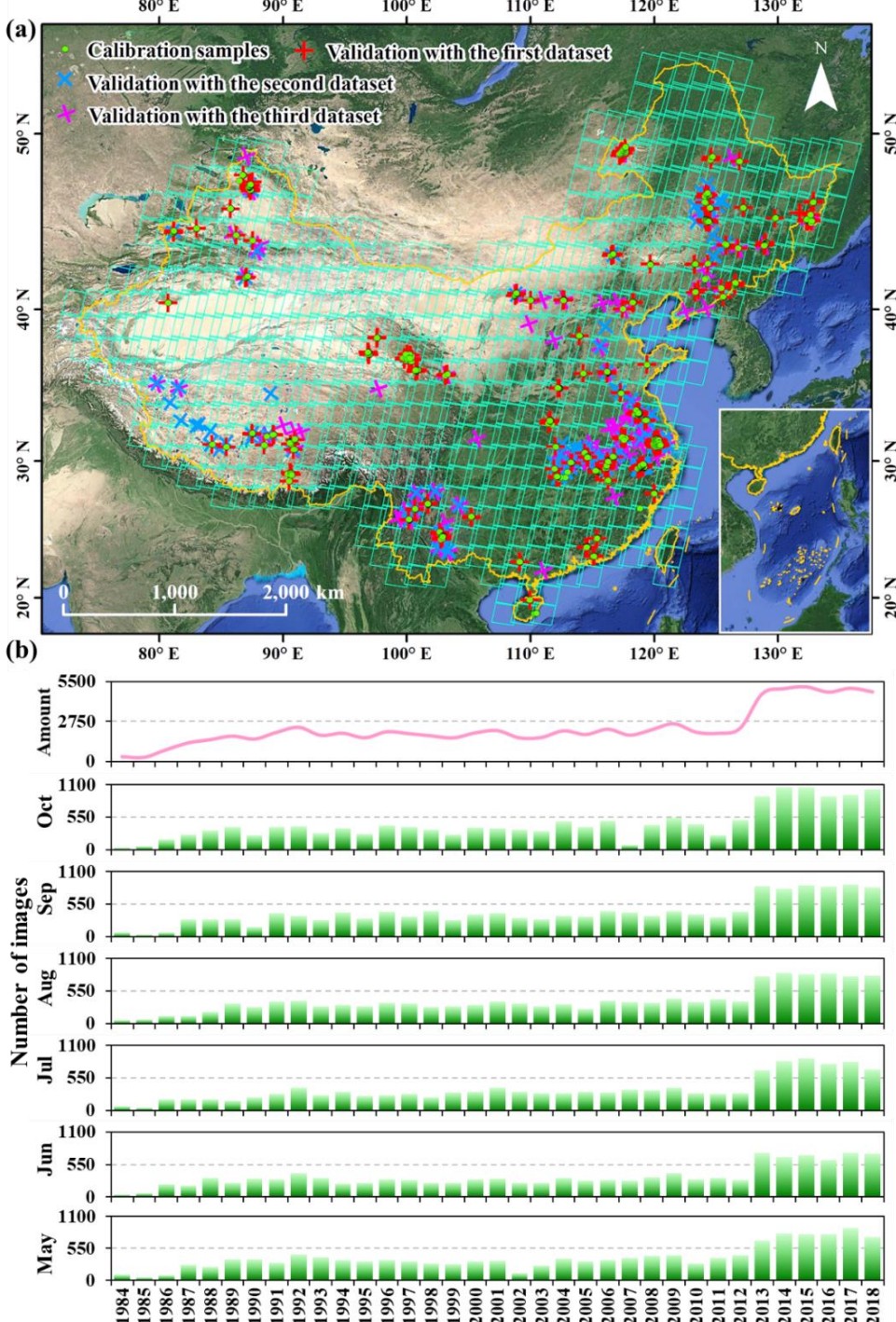

**Figure 2: Location of the sampled waterbodies (lakes or reservoirs) and Landsat Worldwide Reference System 2 (WRS-2) path/row across China (a). Number of Landsat scenes used in ice-free season from 1984 to 2018 (b).**

## 3.5 Statistical analysis

The SDD estimation model performance was assessed using determination coefficient ($R^2$), RMSE, relative RMSE (rRMSE), and mean absolute error (MAE).

$$RMSE = \sqrt{\frac{\sum_{i=1}^{N}(Y_{estimated,i} - Y_{observed,i})^2}{N}}, \tag{3}$$

$$rRMSE = 100 \times \frac{RMSE}{\overline{Y_{observed,\iota}}}, \tag{4}$$

$$MAE = \frac{\sum_{i=1}^{N}|Y_{estimated,i} - Y_{observed,i}|}{N}, \tag{5}$$

where $N$ refers to the number of water samples, $i$ refers to the current water sample number, $Y_{observed,i}$ refers to the in situ SDD measurements, $\overline{Y_{observed,\iota}}$ refers to the average of observed Y, and $Y_{estimated,i}$ refers to the estimated SDD from the Landsat data.

Once the annual mean SDD maps were generated, the average of SDD for each pixel within a lake was calculated for the observation period (1984-2018). For each lake region and individual lake, the spatiotemporal dynamics in SDD were analysed, including the variations of the average, changing trend, number of lakes, and lake surface area. The interannual changing trend was assessed at the 5% significance level, and the slope from linear regression analysis between SDD values and years. These analyses were conducted with the IBM SPSS software. Based on the analysis of interannual change trend in SDD, the lakes in China were divided into three types - lakes with SDD showing significantly increasing (Type I: $p < 0.05$ and slope $> 0$), decreasing (Type II: $p < 0.05$ and slope $< 0$) and non-significant (Type III: $p > 0.05$) trends from 1984-2018.

## 4 Validation of SDD estimation model

The estimation model of lake SDD across China was built using 3/4 of the first matched dataset (976 samples), for which the $R^2$, RMSE, rRMSE, and MAE were 0.79, 100.3 cm, 61.9%, 57.7cm, respectively (Fig.3a). Then, we used 325 samples (1/4 of the first matched dataset) to validate the model, and the validation results indicated stable performance by showing comparative errors ($R^2=0.80$, RMSE = 92.7 cm, RMSE% = 57.6%, MAE= 54.9 cm; Fig.3b). Further, the second and the third datasets were both used to validate model performance with a major focus on testing the temporal transferability of the model (Fig.3c, d). The second dataset (340 samples), collected as part of the Chinese lakes survey conducted by Nanjing Institute of Geography and Limnology, also indicated a good model performance ($R^2=0.78$, RMSE = 74.7 cm, RMSE% = 59.1%, MAE= 42.6 cm; Fig.3c). The third dataset (229 samples) was assembled by the first lake surveys conducted in the 1980s, and was used to validate the model performance for SDD derived from historical remotely sensed data. Our results also demonstrated a stable performance for lake SDD before 1990s ($R^2=0.81$, RMSE = 61.8 cm, RMSE% = 50.6%, MAE= 40.3 cm; Fig.3d). Comparison of validation results for these different periods and datasets demonstrated the stable performance of the SDD model (Fig. 3). Therefore, the estimation of SDD using images acquired by Landsat series of sensors provides a reliable method to examine historical trend in SDD through time series analysis.

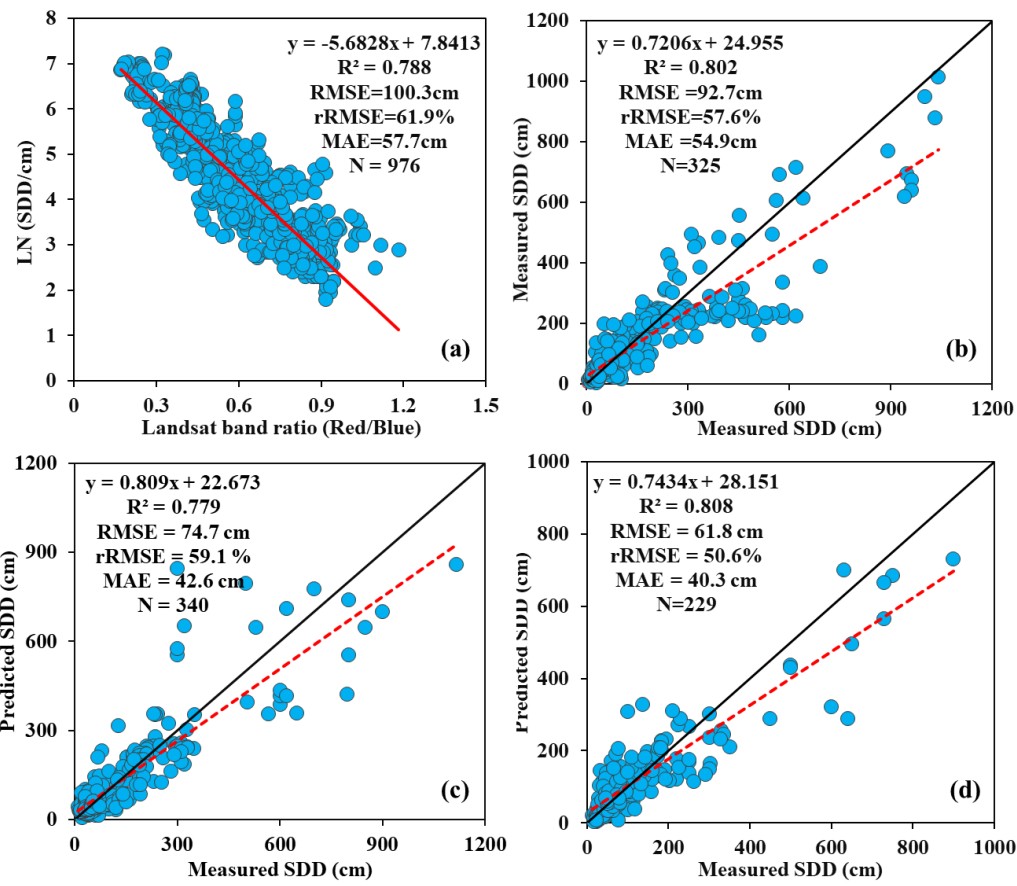

**Figure 3: Model calibration and validation for SDD estimation with Landsat TOA reflectance product acquired by different Landsat sensors, (a) model calibration with 3/4 of the total number of samples from the first dataset, (b) model validated with 1/4 of the total number of samples from the first dataset, (c) model validated with the second dataset independently collected during the limnological survey (2007-2009), and (d) model validated with the third dataset collected in the first lake environmental survey during 1985-1990.**

## 5 Spatial distribution of SDD in lakes in 2018

Fig. 4a shows the spatial distribution of annual mean SDD of lakes across China in 2018, demonstrating remarkable spatial variation, with lakes in the plateau regions generally exhibiting higher SDD than those situated in the flat plain regions. Based on their mean SDD, all lakes across China in 2018 were divided into six levels, i.e., <0.5 m, 0.5-1 m, 1-2 m, 2-3 m, 3-4 m, and >4 m, with 26.4%, 25.7%, 28.8%, 12.5%, 4.3%, and 2.3% of lakes in each SDD level, respectively (Fig. 4b). Although the number of lakes with SDD < 2 m was more numerous (80.9% of lakes), the total area of lakes with SDD of <0.5 m and > 4 m was the largest, accounting for 24% and 24.3% of the total area in each category, respectively (Fig. 4c).

Regarding the annual mean SDD in the five lake regions, the top three regions were TQR (3.37 m), YGR (2.35 m), MXR (1.92 m), followed by ELR (1.50 m) and NLR (0.69 m) (Fig. 4d). Except for the YGR region, lakes with SDD <2 m were most common accounting for 96% (NLR), 82.8% (ELR), 80.5% (MXR) and 77.6% (TQR) of all lakes in the other regions,

respectively (Fig. 4e). In the YGR, the lakes with SDD in 1-3 m range had a wide distribution, and the total proportion of lakes with SDD < 3 m was 85.4% in this region (Fig. 4e). Spatially, the lakes were widely scattered over the ELR, except for the northern and western sections of that region (i.e., northern and southern of Hebei province, northeast of Henan province, northwest of Shandong province and western of Hubei and Hunan provinces). The lakes in the NLR were located in the northwest and southwest of the region. In the YGR, the lakes were clustered in the southern and northeast of the region (i.e., mid-east of Sichuan province and most of Yunnan and Guangxi province). A large number of lakes were inventoried in the TQR, including a collection of large lakes situated in the mid-west and eastern sections of the region, particularly in northwest Tibet and in the western and eastern sections of Qinghai province. In the MXR, the lakes were mainly distributed in the mid-east and mid-west of Inner Mongolia and parts of western and northern of Xinjiang Uygur Autonomous Region.

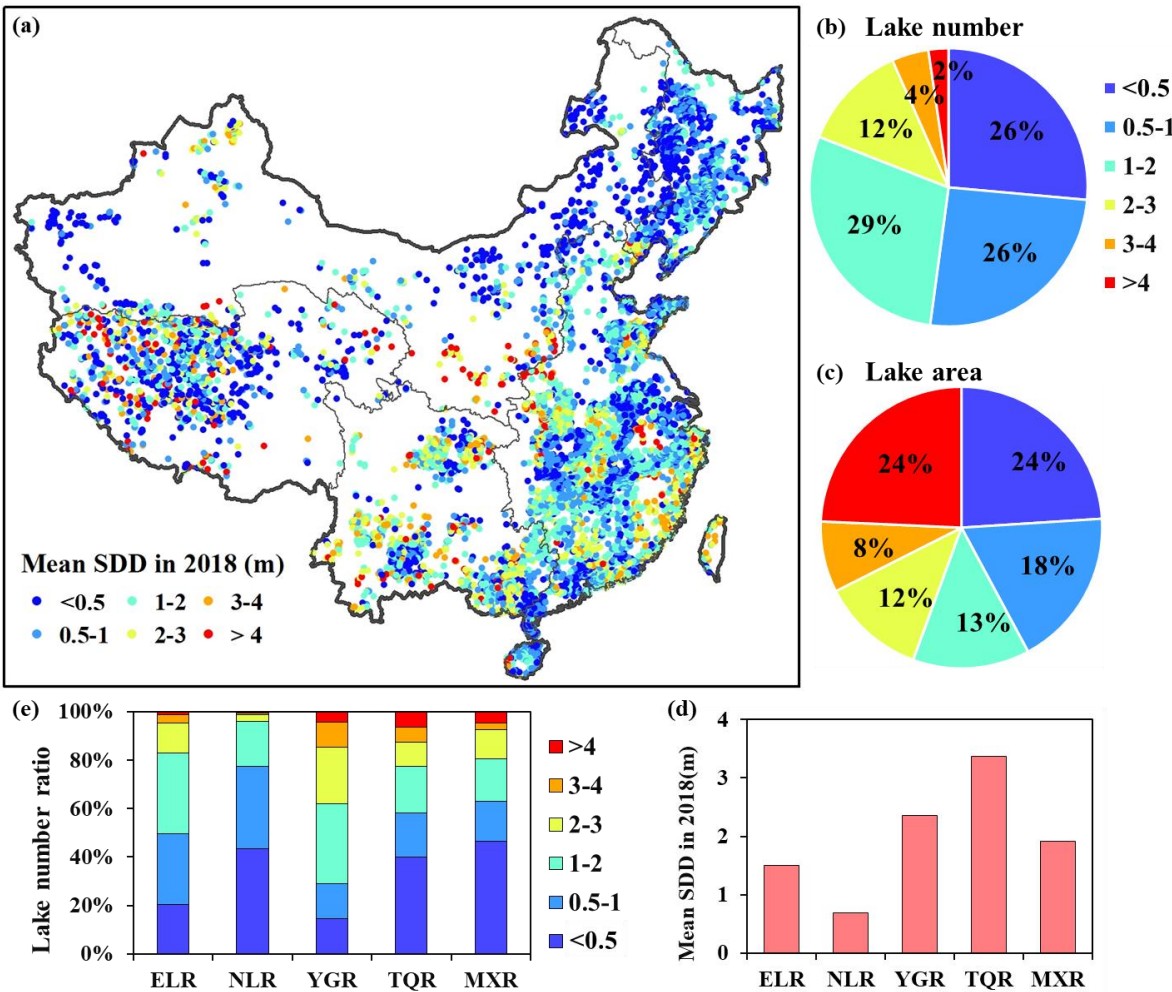

**Figure 4: Annual mean SDD of lakes (>0.01 km² ) across China in 2018. (a) Spatial distribution of lakes with SDD values. (b) Proportion of lake number with SDD values for six levels (i.e., <0.5 m, 0.5-1 m, 1-2 m, 2-3 m, 3-4 m, and >4 m). (c) Proportion of lake area for six SDD levels. (d) Annual mean SDDs in the five lake regions. (e) Proportion of lake number at different SDD levels in the five lake regions.**

**6 Interannual dynamics of lake SDD during 1984-2018**

**6.1 Temporal average and trend in lake SDD**

Similar to the spatial pattern of SDD estimates obtained in 2018, the multi-year average SDD values in each lake region also revealed similar trends, i.e., the lakes located in the plateau region were more transparent than lakes from other
physiographic regions (Fig. 5a). During 1984-2018, the lakes in the NLR exhibited the lowest SDD (mean: 0.60±0.09 m), followed by the ELR (mean: 1.23±0.17 m). The MXR showed intermediate SDD values (mean: 1.63±0.38 m), and the YGR exhibited relatively higher SDD (mean: 2.35±0.21 m). Lakes in the TQP had the clearest water (mean SDD: 3.32±0.38 m; Fig. 5a). As shown in Fig. 5a, mean annual SDD estimates in the five lake regions were in agreement with *in-situ* measured SDD.

Regarding the interannual change trend, with the exception of the TQR, results for the other four lake regions indicated a significant ($p < 0.05$) increasing trend in SDD during the study period (Fig. 5b). At the scale of individual lakes, 55.4% (5,993 out of 10,814) and 3.5% (377 out of 10,814) of lakes experienced statistically significant ($p < 0.05$) increasing and decreasing trends, respectively, and the remaining lakes (41.1%, 4,444 out of 10,814) displayed no significant change (Fig. 5c). Among the five lake regions, except for the MXR, more than half of all lakes exhibited significant increasing trends (Fig. 5c).
Ranked by the total number of lakes exhibiting significant increase in SDD, the lake regions can be ordered as follows: TQR (61.7%, 618 out of 1,002), ELR (57.1%, 3,396 out of 5,943), YGR (54.6%, 829 out of 1,517) and NLR (51.3%, 784 out of 1,528). As for the lakes with decreasing SDD values, the NLR had the highest number of such lakes (8.4%, 128 out of 1,528) followed by the MXR (7%, 58 out of 824) (Fig. 5c).

Among the three types of lakes — lakes with SDD showing significant increasing (Type I), decreasing (Type II) and
280 nonsignificant (Type III) trends from 1984 to 2018, the lake SDDs in the Type I, Type II and Type III were mainly concentrated in 0.5-3 m, 0-2 m and 0-3 m, respectively; the corresponding proportions were 81.11% (4,861 out of 5,993), 80.11% (302 out of 377) and 85.13% (3,783 out of 4,444) of the total number of lakes, respectively (Fig. 5d-f). At the five lake regions scale, regardless of the lake type, the distribution of lake SDDs in the NLR, TQR and MXR appeared similar, while those in the ELR and YGR differed from these three lake regions. The former was mainly distributed in 0-2 m, the
285 latter ranged 0.5-3 m (ELR) and 1-4 m (YGR), respectively (Fig. 5d-f).

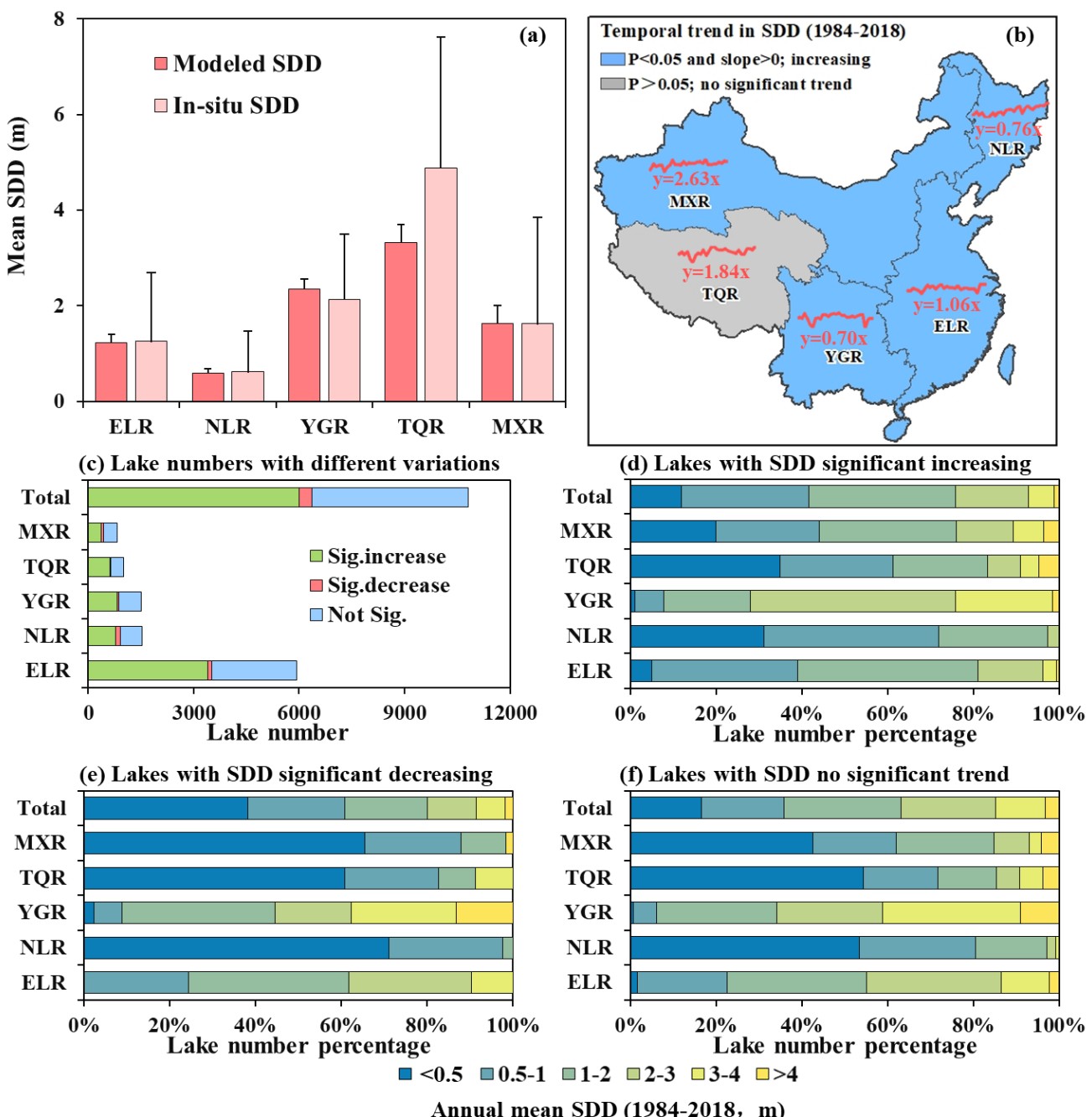

**Figure 5: The interannual dynamics of lake SDDs in China during 1984-2018. (a) Multi-year average SDD values of the modelled and in-situ SDDs in the five lake regions. (b) Interannual trends of mean lake SDDs in five lake regions based on the 5% significant level and slope representing the coefficient of simple linear regression. (c) Number of lakes with SDD showing statistically significant (p < 0.05) increasing (Type I), decreasing (Type II) and nonsignificant (Type III) trends. Proportions of lake numbers with different SDD values (<0.5 m, 0.5-1 m, 1-2 m, 2-3 m, 3-4 m, and >4 m) for: (d) lakes with SDD showing significant increasing trend; (e) lakes with SDD showing significant decreasing trend; and (f) lakes with SDD showing no significant trend.**

## 6.2 Lake SDDs versus lake sizes in China

The annual mean SDD and lake area were both separated into six levels, and the proportions of lakes with different areas in each SDD category are shown in Fig. 6. In terms of the number of different lake areas in the five lake regions, the lakes with annual mean SDD values in the ELR, NLR and YGR were dominated by the area range of 0.01-1 km$^2$, followed by that of 1-10 km$^2$. In the MXR, the lakes were mainly dominated by the area range of 1-10 km$^2$, followed by that of 0.01-1 km$^2$ (Fig. 6a-f). In the TQR, when the SDDs < 2 m, the lakes covering the area range of 1-10 km$^2$ were in the majority (Fig. 6a-c); when the SDDs > 2 m, the lakes with the area range >10 km$^2$ occupied a dominant position, especially for lakes with the area range of 10-50 km$^2$ and 100-500 km$^2$ (Fig. 6d-f).

Among the three types of lakes in each SDD category, there is a similarity in the distribution of lakes with different sizes between the Type I and Type III, while that of Type II differentiated from these two types of lakes (Fig. 6). In the ELR, NLR and YGR, almost more than 50% of the lakes ranged 0.01-1 km$^2$ among the lakes of Type I and Type III. The lakes of Type II, located in the three lake regions, with SDD values of 0.5-1 m in the ELR, and of <0.5 m and 2-3 m in the NLR were dominated by the area size of 1-10 km$^2$, while the remaining lakes were mostly with the area range of 0.01-1 km$^2$ (Fig. 6a-f). In the MXR, the number of lakes covering the area range of 1-10 km$^2$ in the three types of lakes was much larger than that of other sizes among the lakes with SDDs in the range 0-3 m (Fig. 6a-d). When the lake SDDs were > 3 m in this lake region, most of the three types of lakes were dominated by the lakes covering the area range of 0.01-1 km$^2$, apart from the lakes of Type III with SDD values > 4 m that the proportion of lakes with the area range of 1-10 km$^2$ was slightly higher than that with the area range of 0.01-1 km$^2$, (Fig. 6e-f).

The distribution of the three types of lakes with different lake sizes in the TQR differed from those in the other four lake regions. For the lakes of Type I and Type III in the TQR, when the SDDs ranged 0-2 m, the proportions of lakes covering the area range of 1-10 km$^2$ were the largest, somewhere between 49.64% and 81.12% (Fig. 6a-c). When the SDDs ranged 2-3 m, the lakes with the area range of 10-50 km$^2$ in the Type I and that of 100-500 km$^2$ in the Type III had the largest proportions of numbers, accounting for 40.43% and 35.00%, respectively (Fig. 6d). When the SDDs exceeded 3 m, the lakes covering the area range of 100-500 km$^2$ were dominant in the two types of lakes, followed by the area range of 10-50 km$^2$ (Fig. 6e-f). For the lakes of Type II in the TQR, the lakes with SDDs in the <0.5 m category were distributed in the area range of 10-50 km$^2$, followed by that of 50-100 km$^2$ (Fig. 6a). When SDDs were in the 0.5-1 m category, the numbers of lakes with the area range of 1-10 km$^2$ and 10-50 km$^2$ were the largest, where the corresponding percentages were 40.00% (Fig. 6b). When SDDs were in the 1-2 m category, there were two kinds of lakes whose areas were in the range of 0.01-1 km$^2$ and 50-100 km$^2$, and their numbers were the same (Fig. 6c). When SDDs were in the 3-4 m category, only the lakes with the area range of 1-10 km$^2$ existed (Fig. 6e).

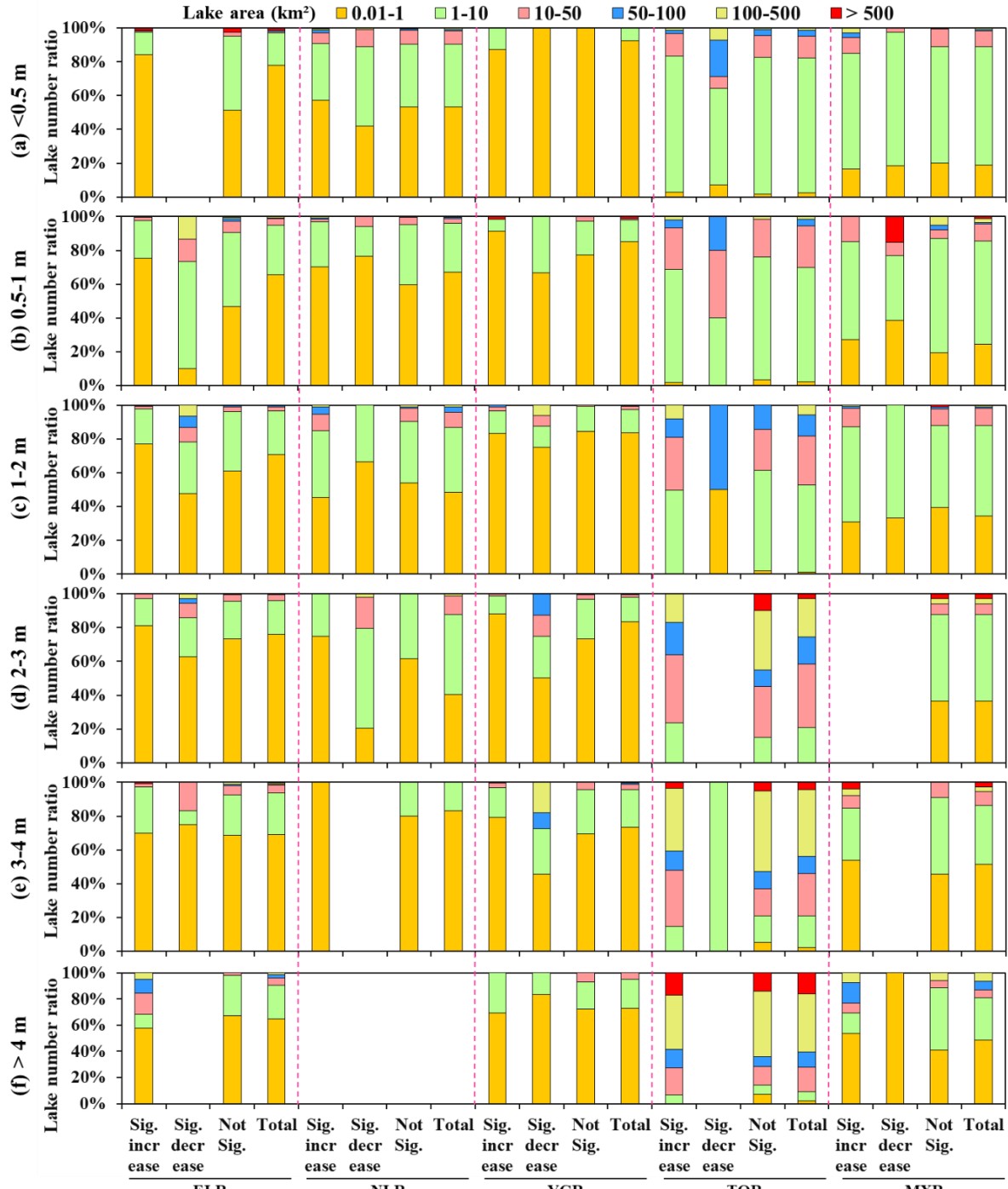

**Figure 6: Proportions of lake numbers in different areas in the six SDD categories. The six SDD categories are: (a) <0.5 m; (b) 0.5-1 m; (c) 1-2 m; (d) 2-3 m; (e) 3-4 m; (f) >4 m. The SDD values are the average of estimated results in each lake during 1984-2018. In the five lake regions, the lakes are further divided into three types — lakes with SDD showing significant increasing (Type I), decreasing (Type II) and nonsignificant (Type III) trends during 1984-2018.**

## 6.3 Spatial distribution of lakes with different SDD values

The spatial distributions of lakes and their number of lakes and areas of the three types of lakes in five lake regions are presented in the Fig.7. In the SDD of <0.5 m category (Fig.7a), the NLR had the largest lake numbers and areas in the three types of lakes, accounting for 34.51% and 33.20% in Type I, 63.19% and 48.17% in Type II and 44.46% and 34.38% in Type III of the number of lakes and areas in the lake region, respectively. Spatially, the lakes in Type I and Type III were mainly distributed in the central section of the ELR, the western section of the NLR, the mid-west of the TQR and the mid-east of the MXR, while those in Type II were concentrated on the western section of the NLR and eastern section of the MXR.

In the SDD of 0.5-3 m categories (Fig.7b-d), the lakes of Type I and Type III were the most in the ELR, but the largest total lake areas of the five lake regions were different from these two types of lakes. Specifically, in the lakes of Type I, the total lake areas in the TQR were the largest, with the percentages of 36.38% (SDD: 0.5-1m), 44.14% (SDD: 1-2 m) and 61.03% (SDD: 2-3 m), respectively (Fig.7b-d). Regarding the lakes of Type III, the ELR (TQR) had the largest proportion of lake area when SDD was 0.5-2 m (2-3 m). The percentages of lake area when SDD was 0.5-2 m in the ELR were 76.80% (SDD: 0.5-1m) and 46.90% (SDD: 1-2 m), while that in the TQR was 46.65% (SDD: 2-3 m) (Fig.7b-d). For the lakes of Type II, the region that had the largest proportions of lake number and area was inconsistent in each SDD category (0.5-3) m. When the SDDs were in the range of 0.5-1 m, the NLR had the largest lake number, while the MXR had the highest percentage of lake area (Fig.7b). When the SDDs ranged from 1-2 m, the number of lakes and area in the ELR were the largest (Fig.7c). When the SDDs were around 2-3 m, the lake number in the NLR was the largest and the total lake area in the ELR was the largest (Fig.7d). Spatially, lake distributions of the Type I and Type III with SDD range of 0.5-2 m were concentrated on the most places of the ELR, the northwest and southeast of the NLR, the southern section of the YGR, the mid-west of the TQR, and the mid-east and the northern section of the MXR (Fig.7b-c). When these two types of lakes SDD were in the range of 2-3 m, they were distributed in the central and southeast coast of the ELR, the central and southwest of the YGR, and the western section of the TQR (Fig.7d). For the Type II of lakes with SDD falling in the range 0.5-3 m, their distributions were scattered over part of the central and southeast coast of the ELR, and southwest of the YGR (Fig.7b-d).

In the SDD of 3-4 m category (Fig.7e), the regions that had the most lakes in the three types of lakes were the YGR (Type I: 53.56%), ELR (Type II: 48.00%), and ELR (Type III: 53.19%), respectively. The regions that had the largest lake area were the TQR (Type I: 63.51%), YGR (Type II: 90.06%), and TQR (Type III: 75.22%), respectively. Spatially, the lakes of Type I and Type III were concentrated at the junction of the ELR, YGR and MXR; the southeast coast of the ELR; the southern section of the YGR; and the western section of the TQR. The lakes of Type III were mainly distributed in the part of the southeast coast of the ELR and the southern section of the YGR.

Regarding the SDD of >4 m category (Fig.7f), the TQR had the largest lake number and area in the lakes of Type I, accounting for 39.19% of the number of lakes and 87.34% of the total lake area. For the lakes of Type II, a few lakes existed in the MXR and YGR. For the lakes of Type III, the YGR had the most lakes and the TQR had the largest total lake area,

accounting for 40.28% of the number of lakes and 87.00% of the total lake area, respectively. Spatially, the distributions of these lakes were similar to the lakes with SDD range of 3-4 m range.

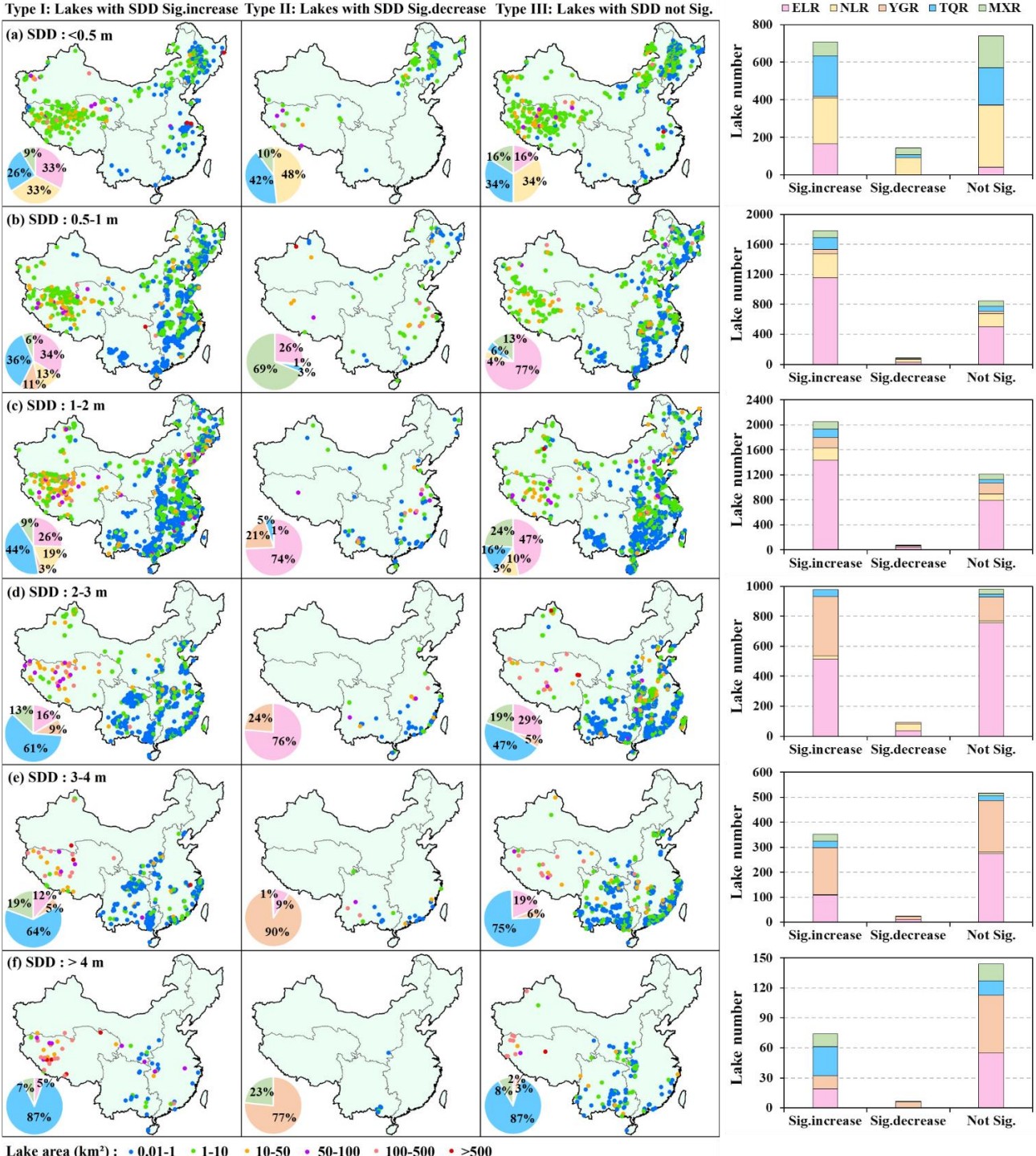

**Figure 7: Spatial distribution of lakes with multi-year average SDD values during 1984-2018. The SDD values were divided into six levels: (a) <0.5 m; (b) 0.5-1 m; (c) 1-2 m; (d) 2-3 m; (e) 3-4 m; (f) >4 m. The lakes were separated into three types of lakes—lakes with SDD showing significant increasing (Type I), decreasing (Type II) and nonsignificant (Type III) trends during 1984-2018. Proportions of total lake area and lake number in each lake region are shown in the pie charts and histogram, respectively.**

## 7 Comparison with past studies and uncertainties

Several past studies have examined the spatiotemporal variation of SDD in lakes across China (or parts of China), but these investigations were mainly based on MODIS images to estimate SDD in large lakes (>10 $km^2$) and primarily focused on the period after 2000 (Feng et al., 2019a; Liu et al., 2020a; Pi et al., 2020; Wang et al., 2020a). Therefore, it becomes a challenge to compare these past results with the results of the present study due to difference in the period of interest, resolution of the satellite images and lake size (> 0.01 km² in our study). Zhang et al. (2021) adopted an empirical model to retrieve SDD of lakes (>10 $km^2$) across China based on Landsat surface reflectance products (2016 - 2018) within GEE. Because of the similarity of methods and images used in Zhang et al. (2021) and the present study, it provides a unique opportunity to compare lake SDD estimation models across China proposed by these two researches. To that end, we used available *in-situ* SDD data (2019 – 2020) collected at monitoring stations in Lake Taihu and Lake Dianchi to assess the accuracy of the two models. As shown in Fig. 8 and demonstrated by statistical parameters (higher $R^2$, lower RMSE, rRMSE and MAE), the estimation model proposed by our study exhibited better performance to retrieve SDD in both Lake Taihu (Fig.8c) and Lake Dianchi (Fig. 8d).

Whilst previous studies have demonstrated the application of Landsat series data (5 TM/ 7 ETM+/ 8 OLI) with the proposed model can provide accurate long-term coverage of SDD of lakes in China (Zhang et al., 2021; Song et al., 2020; Deutsch et al., 2018; Bonansea et al., 2015; Mccullough et al., 2013), several systemic errors on SDD results could not be avoided. On the one hand, the SDD estimation model proposed in this study contained some errors, where the model validation yielded these results: $R^2$=0.80, RMSE = 92.7 cm, RMSE% = 57.6%, MAE= 54.9 cm. On the other hand, different atmospheric correction methods can cause diverse effects on the Landsat images (Bonansea et al., 2015; Lee et al., 2016). The calibrated TOA reflectance products within the GEE were produced using the equations developed by Chander et al. (2009). Nevertheless, these systemic errors do not significantly affect the overall trends of SDD of lakes in China (Bonansea et al., 2015; Deutsch et al., 2018; Zhang et al., 2021). In addition, under the influence of climate change or human activities, such as floods and droughts, urbanization, and farmland reclamation, the boundaries for some small lakes (< 1 $km^2$) may vary greatly, which could cause the uncertainty of SDD estimation (Yang et al., 2021; Zhang et al., 2019). This is a limitation of the assumption for small lakes with static boundaries. In the future, further research on the relationship between the area of small lakes and the accuracy of SDD simulation would aid in addressing this limitation.

## 8 Data availability

The dataset of water clarity of lakes developed in this study consists of one shapefile containing the annual mean values of water clarity in each lake (size > 0.01 km³) during 1990-2018, with a temporal resolution of 5-year. The dataset can now be accessed through the website of the National Tibetan Plateau Data Center (http://data.tpdc.ac.cn): DOI: 10.11888/Hydro.tpdc.271571.

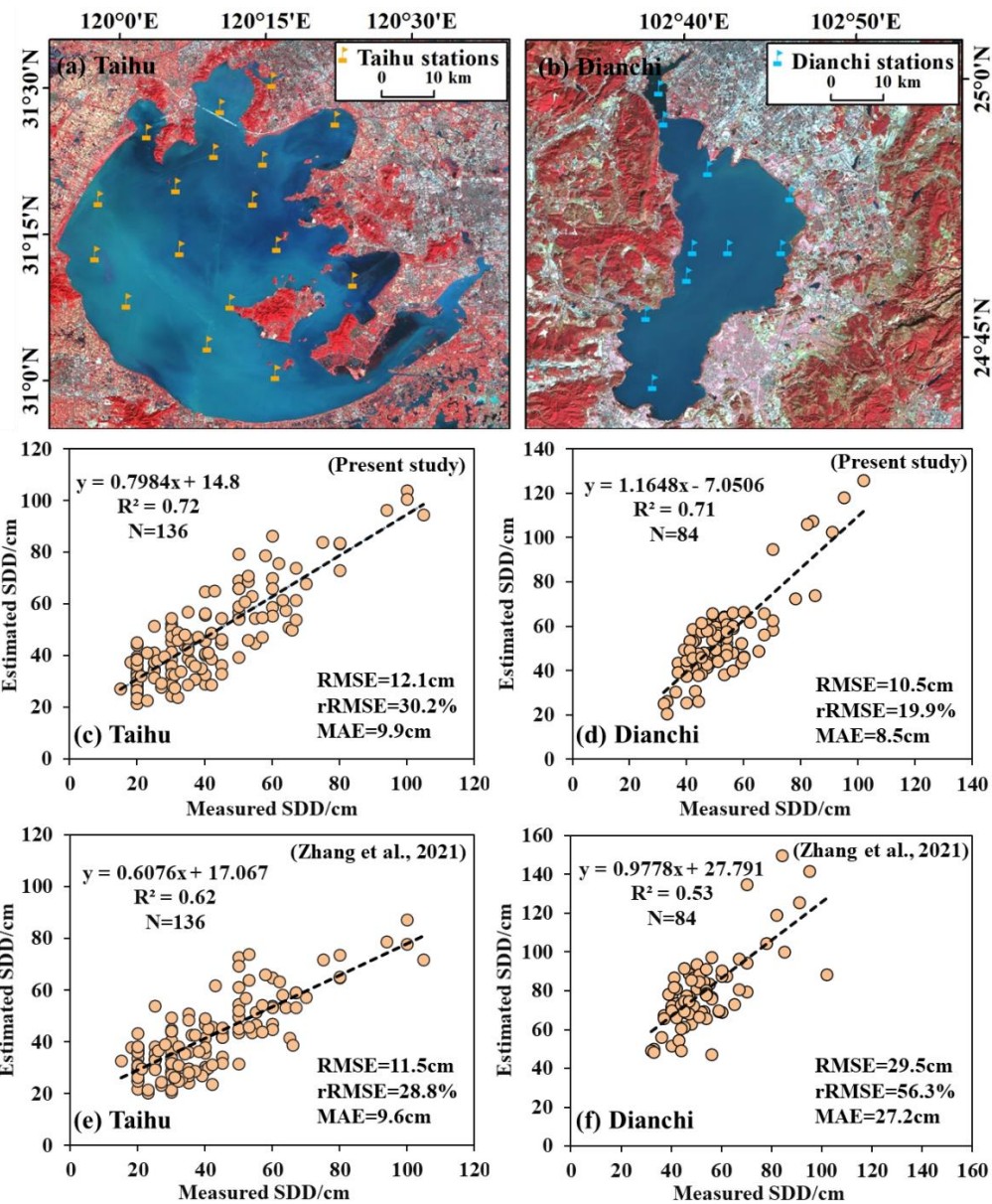

**Figure 8: Comparison of different SDD estimation models based on Landsat images within the GEE. (a and b) Spatial distribution of monitoring stations located in lake Taihu and lake Dianchi, respectively. (c – f) The regression line between the measured SDD in lake Taihu (N = 136) and lake Dianchi (N = 84) during 2019 – 2020 and estimated SDD values that were obtained from the estimation models developed in this study and Zhang et al. (2021), respectively.**

## 9 Conclusions

As a comprehensive indicator of water eutrophication, encompassing nutrient enrichment, algal abundance and suspended sediment, water clarity can serve as a valuable index for tracking the ecological health of aquatic ecosystems and guiding the actions of water resources managers. Although field measurement of water clarity can easily be made with a Secchi disk apparatus, this approach is not suitable for long-term time series measurements of lake water clarity at regional and national scales. This information is highly valuable, and can be extracted from archived satellite data. *In-situ* water clarity data collected in lakes across China during 2004-2018 was used to calibrate and validate SDD models that incorporate top of atmosphere reflectance product and Google Earth Engine to map the spatiotemporal dynamics of SDD over a 35-year time span (1984-2018). The SDD model was validated using different datasets, and results confirmed the stable performance and temporal transferability of the SDD estimation model. Derived SDD estimates were analysed at the lake region and at the individual lake scales. During the study period (1984-2018), annual mean SDD values in the TQR, YGR, MXR, ELR and NLR regions were 3.32±0.38 m, 2.35±0.21 m, 1.63±0.38 m, 1.23±0.17 m and 0.60±0.09 m, respectively. Among the 10,814 lakes with >10 years of SDD results, 55.4% and 3.5% experienced statistically significant ($p<0.05$) increasing and decreasing trends of water clarity, respectively. The remaining lakes (41.1%) displayed no significant trends. With the exception of the MXR, more than half of lakes in all the other regions exhibited a significant trend of increasing water clarity. In the ELR, NLR and YGR regions, most of the lakes displaying either an increase or decrease in SDD tended to be of 0.01-1 $km^2$ in size whereas in the TQR and MXR lakes exhibiting clear trends in SDD were mostly large lakes (>10 $km^2$). Spatially, the lakes in the plateau regions (TQR, YGR) generally exhibited higher SDD than those situated in the flat plain region. The time series of water clarity information presented in this study could aid local, regional and national decision-making on policies and management for protecting/improving inland water quality in China. The research approach implemented also could potentially be used to map water clarity in lakes at the global scale, an effort that can provide useful information for evaluating decadal trends in surface water quality resulting from adoption of pollution control policies.

**Author contributions.** KS, HT, GL, and HD designed the study; KS, GL, and HT performed the research; KS, HT, GL, QW, ZW, CF, YD, ZW, DL, KS, BZ, and XW collected and analysed the data; and KS, HT, GL, TK, PJ, and HD wrote the paper. All authors contributed to the interpretation of findings, helped revise the manuscript, and approved the final manuscript for submission.

**Competing interests.** The authors declare that they have no conflict of interest.

**Acknowledgements.** This research was jointly supported by the Natural Science Foundation of China (41730104, 42001311, 42171385), China postdoctoral science foundation (2020M681056), Strategic Priority Research Program of the Chinese Academy of Sciences (XDA19070501), Research instrument and equipment development project of Chinese Academy of Sciences (YJKYYQ20190044), and National Earth System Science Data Center, China (www.geodata.cn). The authors wish to thank Dr. Ying Zhao, Jianhang Ma, and Ming Wang for their capable assistance in the field sampling and laboratory measurements.

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
