# Peer review of "A Landsat derived annual inland water clarity dataset of China between 1984 and 2018"

_Earth System Science Data, 2021_

## Referee Comment (RC2)

[referee-annotated manuscript omitted]

---

## Author Response (AR1)

Response to Reviewer comments
Manuscript Number: essd-2021-227-RC1
Manuscript Title: Water clarity annual dynamics (1984–2018) dataset across China derived from Landsat images in Google Earth Engine

Response to anonymous referee #1:
Anonymous referee #1:
In this study, the authors developed / provided a valuable water clarity data set across China during 1984-2018 from Landsat images by GEE platform. This data set was validated, and spatio-temporal patterns of water clarity were also analyzed. Overall, this manuscript is written well and suitable to publish in ESSD. I recommend a minor revision based on the comments below to improve the quality of this manuscript / data set before publication.

**Major comments:**
1. The structure of Abstract is not clear. From beginning of data set development, validation, to spatio-temporal pattern… could be better.

Response: Thank you for this suggestion, we have adjusted part of the structure of Abstract, and the detailed revision can be seen below.

Water clarity provides a sensitive tool to examine spatial pattern and historical trend in lakes trophic status. Yet, this metric has insufficiently been explored despite the availability of remotely-sensed data, especially for long-term monitoring. Therefore, we utilized Landsat top of atmosphere reflectance products within Google Earth Engine in the period of 1984-2018 to retrieve the average SDD for each lake in each year. Three Secchi disk depth (SDD) datasets were used for model calibration and validation from different field campaigns mainly conducted during 2004-2018. The red/blue band ratio algorithm was applied to map SDD of lakes (> 0.01 km²) based on the first SDD dataset, where $R^2 = 0.79$, rRMSE = 61.9%. The other two datasets were used to validate the temporal transferability of SDD estimation model, which were indicated the model had a stable performance. The spatiotemporal dynamics of SDD were analyzed at the five lake regions and individual lake scales, and the average, changing trend, lake number and area, and spatial distribution of lake SDDs across China were presented. In 2018, we found that the lakes with SDDs < 2 m accounted for the largest proportion (80.93%) of the total lakes, but the total area of lakes with SDD between 0-0.5 m and > 4 m were the largest, accounting for 48.28% of the total lakes. During 1984-2018, lakes in the Tibetan-Qinghai Plateau lake region (TQR) had the clearest water with an average value of 3.32±0.38 m, while that in the Northeastern lake region (NLR) exhibited the lowest SDD (mean: 0.60±0.09 m). Among the 10,814 lakes with SDD results more than 10 years, 55.42% and 3.49% of lakes experienced significant increasing and decreasing trends, respectively. At the five lake regions, except for the Inner Mongolia-Xinjiang lake region (MXR), more than half of the total lakes in every other lake region exhibited significant increasing trends. In the Eastern lake region (ELR), NLR and Yungui Plateau lake region (YGR), almost more than 50% of the lakes that displayed an increase or decrease in SDD

were mainly distributed in an area of 0.01-1 km$^2$, whereas that in the TQR and MXR were primarily concentrated in large lakes (> 10 km$^2$). Spatially, lakes located in the plateau regions generally exhibited higher SDD than those situated in the flat plain regions. The dataset can now be accessed through the website of the National Tibetan Plateau Data Center (http://data.tpdc.ac.cn): DOI: 10.11888/Hydro.tpdc.271571.

2. The authors mapped the spatiotemporal variation of SDD in lakes (>1 ha) across China from 1984 to 2018. How the lakes are mapped properly and accurately in this study? I think GEE has limitation in conducting this. As the cloud and shadow effects on lake boundaries, an automatic / semi-automatic method is not possible to map lakes accurately. In addition, in the middle and lower reaches of Yangtze River regions, the lake boundaries are very difficultly differentiated from other water /non-water classifications. How the authors do these? The lake boundaries were examined with origin Landsat images? How the seasonal inconsistency for data selection was considered? How the rivers and reservoirs are excluded from water bodies? The authors compared the results of mapped lakes with existing lake data set in China? This is necessary for validation the accuracy of lake mapping for this study. The very small size lakes are included. The land contamination to lake water was considered?

Response: Thank you for these comments, it is helpful for us to improve the quality of this paper. According to your questions, we have made some explanations below.

**1) As for the question about lake boundaries.**
First of all, we are sorry that the extraction of lake boundaries in present study was not described clearly, which makes the reviewer more confused about it. The revised content could be seen in the section of "3.1 Waterbody mask" of revised-manuscript. Next, we will explain it in detail.
Based on the previous study by Song et al. (2020), the lake boundaries (lakes and reservoirs) with an area > 0.01 km$^2$ across China were derived from Landsat 8 OLI images mainly acquired in 2016, while some images in 2014, 2015, 2017, or 2018 were used when images in 2016 were unavailable due to cloud or haze contamination. Detailed description of extracting boundaries could be seen in the research of Song et al. (2020). The Figure 1 shows the result of using these lake boundaries to map SDD at a national scale with OLI images mainly acquired in 2016 (Song et al., 2020).

It is well-known that some lakes in China are changing greatly over time. On the basis of the lake boundaries derived from the study of Song et al. (2020), we dealt with these changing lakes separately to obtain their boundaries in each year during the period of 1984-2018. We mainly referred to the research of Zhang et al. (2019) to obtain the information of which lake boundary has changed and what year the lake started to vary. This research examined multi-decadal lake area changes in China during 1960s–2015, using historical topographic maps and Landsat satellite images, including lakes as fine as ≥1km$^2$ in size. The datasets of lake boundaries (1960s-2020) have been published on the National Tibetan Plateau Data Center. The Figure 2 displays the spatial variation of lake boundaries from 1990 to 2015 with a temporal resolution of 5 years. As for the reservoirs, we mainly viewed the Landsat (5/7/8)

images to confirm the changing region. With respect to the small lakes with an area < 1km², we assumed that their boundaries didn't change during the study period.

We delineated boundaries of these changing lakes using Landsat images during 1984-2018. The cloudless TOA image of each path and row was downloaded through GEE platform, processed to derive the Modified Normalized Difference Water Index (MNDWI) as follows:

$$MNDWI = (R_{rc,Green} - R_{rc,SWIR})/(R_{rc,Green} - R_{rc,SWIR}) \qquad (1)$$

where, $R_{rc,Green}$, $R_{rc,SWIR}$ is the Rayleigh scattering reflectance in the green band, and short-wave infrared (SWIR) band, respectively. First, we used MNDWI, combined with Tasseled Cap Transformation (TC) and a density slicing with multi-threshold approach, to build a decision tree for retrieving water body boundaries using the ENVI software package (Rokni et al., 2014; Xu et al., 2006). Then, Landsat images acquired during 1984-2018 were classified into water and non-water areas (Feyisa et al., 2014). The extracted water bodies were subsequently converted into polygons with contiguous pixels and stored in shape file format using the ArcGIS10.4 (ESRI Inc. Redlands, CA, USA). We divided water bodies into lakes, reservoirs, and rivers according to their shoreline features, and also through referencing to the Global Reservoirs and Dams database (Lehner et al., 2011), Chinese Reservoirs and Dams database, and high-resolution images from Google Earth to tell rivers and reservoirs from water bodies.

Jensen. (2006) pointed out that the various surface objects have different reflectance to NIR band. For instance, the NIR band can be largely reflected by land and vegetation and strongly absorbed by water, which leads to a stark contrast between the land and water reflectance, especially for the shallow lakes or reservoirs. The problem of land contamination to water is still a challenge for retrieving water quality parameters precisely (Jensen. 2006; Hou et al., 2018). In our study, in order to avoid the influence of adjacent land on water bodies, one pixel buffer inward of water boundary was removed for lakes with an area ≤ 1 km², and two pixels for lakes with an area > 1 km². This method has been demonstrated to be effective in other studies related to SDD. For example, Liu et al. (2020) and Wang et al. (2020) excluded two pixels and one pixel from large lake boundaries (> 20 km²) extracted by MODIS images with a spatial resolution of 500 m.

[Figure]

Figure 1: The SDD spatial variation of lakes in China with OLI images mainly acquired in 2016.
Note: this figure derived from the study result of Song et al. (2020).

[Figure]

Figure 2: The spatial variation of lake boundaries from 1990 to 2015 with a time resolution of 5 years. Note: the data sets used in this figure sourced from the study of Zhang et al. (2019).

**2) Aa for the question about "How the seasonal inconsistency for data selection was considered?"**

In the manuscript of this study, we may not describe this part clearly, here, we have given the detailed explanation. Based on the GEE platform, the TOA images were mainly collected during the ice-free season (May to October) from 1984 to 2018 in the TQR, MXR, NLR. In order to ensure the consistency of images used in the five lake regions, the TOA images in the ELR and YGR were also designed to select from May to October in each year. However, the image dates in the YGR were actually from January to December due to lack of good-quality images in the selected period. The purpose of this paper was to research the interannual variations of lakes water clarity during 1984-2018, so the TOA images used to estimate water clarity of lakes were mainly from May to October. Despite the images used in the YGR were slightly different from the other four lake regions, it would cause little impact on the analysis of interannual variations in water clarity of lakes across China. Zhang et al. (2021) applied Landsat 8 images in the nonfreezing period from June to October during 2016-2018 to map spatial distribution of the SDD across China and calculated the average SDD for each lake on the basis of the estimated SDD values from 2016 to

2018. In addition, many studies have demonstrated annual mean SDD values for different lakes across China through computing the results of different months within a year (Liu et al., 2020; Pi et al., 2020; Feng et al., 2019). The revised content could be seen in the section of "3.3 Acquisition and processing of Landsat imagery data" of revised-manuscript.

**3) As for question about "The authors compared the results of mapped lakes with existing lake data set in China? This is necessary for validation the accuracy of lake mapping for this study."**

We have considered this question seriously. In the seventh part of this study, the reason has been given why we chose to compare the SDD estimation models proposed by Zhang et al. (2021) and in our study. The result of comparison showed that the estimation model built by our study exhibited better performance to retrieve SDD in both examined lakes (Taihu and Dianchi). Here, we make a comparison with the results of estimated SDD of lakes across China acquired from Zhang et al. (2021) and the present study. Zhang et al. (2021) applied Landsat 8 images in the nonfreezing period during 2016-2018 to map spatial distribution of the SDD across China and calculated the average SDD for each lake (n=641; size $\geq$ 10km$^2$) on the basis of the estimated SDD values from 2016 to 2018. In order to make the results more comparable, we first obtained the lakes (n=639) with an area $\geq$ 10km$^2$, excluding the reservoirs. Then, the average SDD of these lakes were calculated based on the estimated SDD values from 2016 to 2018. The Figure 3 shows the distribution of the average SDD for each lake and five lake regions across China in the period of 2016–2018 in present study. The Figure 4 displays the results from the study of Zhang et al. (2021). Comparing these two Figures, the spatial distributions of the average SDD for each lake in the MXR, NLR, ELR, and YGR demonstrated in present study (Figure 3.b-e) are similar to the results (Figure 4.b-e) of Zhang et al. (2021), while the mean values of lake SDDs in the TQR (Figure 3.f) are a little higher than that (Figure 4.f) showed in the study of Zhang et al. (2021). As for the average of estimated SDD for five lake regions, regional distribution in present study (Figure 3.g) is as follows (in decreasing order): TQR > YGR > MXR > ELR > NLR which is consistent with the distribution of in-situ measured SDD in these lake regions, while that (Figure 4.g) in the study of Zhang et al. (2021) is: YGR > TQR > MXR > ELR > NLR. With respect to the number of lakes in different categories of SDD in the period of 2016–2018, the distribution pattern in the two studies is similarity, though the quantity is slightly differences in some categories.

[Figure]

Figure 3: The distribution of the SDD of lakes (≥ 10km²) in the period of 2016–2018. (a) Geographical location of the five lake regions. The spatial distribution of the average SDD for each lake in the MXR (b), NLR (c), ELR (d), YGR (e), and TQR (f). (g) The purple histogram shows the average SDD for each lake zone and all lakes in China during 2016-2018 (number of lakes: TQR (367), YGR (14), MXR (87), ELR (118), and NLR (53)), while the bright pink histogram displays the mean values of in-situ measurement SDDs in the five lake regions and total lakes in China during 2015-2019 (number of samplings: TQR (102), YGR (73), MXR (177), ELR (351), and NLR (135)) due to the quantity of water samplings distributed across China from 2016 to 2018 is a little small. (h) The color histogram reveals the number of lakes in five categories of SDD ((0–0.5] m, (0.5–1] m, (1–2] m, (2–4] and (4–9) m) in the period of 2016–2018.

[Figure]

Figure 4: The lakes in China divided into five lake zones (a) according to Ma et al. [17]. The spatial distribution of the SDD in the period of 2016–2018 in IMXL (b), NPL(c), EPL (d), YGPL (e), and TPL (f) lakes. The gray histogram plots show the average SDD for each lake zone and all lakes in China (g), where n is the number of lakes in each individual dataset. The color histogram plots show the number of lakes with water area larger than 10 km 2 in four categories of SDD ((0–0.5] m, (0.5–1] m, (1–2] m and (2–9) m) in the period of 2016–2018 (h). Note: this figure derived from the study result of Zhang et al. (2021).

**References:**

Zhang, Y., Zhang, Y., Shi, K., Zhou, Y.,Li, N.: Remote sensing estimation of water clarity for various lakes in China, Water Res., 192, 116844-116844, doi:10.1016/j.watres.2021.116844, **2021**.

Liu, D., Duan, H., Loiselle, S., Hu, C., Zhang, G., Li, J., Yang, H., Thompson, J. R., Cao, Z., Shen, M., Ma, R., Zhang, M.,Han, W.: Observations of water transparency in China's lakes from space, Int. J. Appl. Earth Obs. Geoinf. , 92, 102187, doi:10.1016/j.jag.2020.102187, **2020**.

Pi, X., Feng, L., Li, W., Zhao, D., Kuang, X.,Li, J.: Water clarity changes in 64 large alpine lakes on the Tibetan Plateau and the potential responses to lake expansion, ISPRS J. Photogramm. Remote Sens. , 170, 192-204, doi:10.1016/j.isprsjprs.2020.10.014, **2020**.

Song, K., Liu, G., Wang, Q., Wen, Z., Lyu, L., Du, Y., Sha, L.,Fang, C.: Quantification of lake clarity in China using Landsat OLI imagery data, *Remote Sens. Environ., 243*, 111800, doi:10.1016/j.rse.2020.111800, **2020**.

Feng, L., Hou, X. J.,Zheng, Y.: Monitoring and understanding the water transparency changes of fifty large lakes on the Yangtze Plain based on long-term MODIS observations, Remote Sens. Environ., 221, 675-686, doi:10.1016/j.rse.2018.12.007, **2019**.

Zhang, G., Yao, T., Chen, W., Zheng, G., Shum, C. K., Yang, K., Piao, S., Sheng, Y., Yi, S., Li, J., O'Reilly, C. M., Qi, S., Shen, S. S. P., Zhang, H.,Jia, Y.: Regional differences of lake evolution across China during 1960s–2015 and its natural and anthropogenic causes, Remote Sens. Environ., 221, 386-404, doi:10.1016/j.rse.2018.11.038, **2019**.

Zhang, G.: China lake dataset (1960s-2020). National Tibetan Plateau Data Center, DOI: 10.11888/Hydro.tpdc.270302. CSTR: 18406.11.Hydro.tpdc.270302, **2019**.

Feyisa, G. L., Meilby, H., Fensholt, R.,Proud, S. R.: Automated Water Extraction Index: A new technique for surface water mapping using Landsat imagery, Remote Sens. Environ., 140, 23-35, doi:10.1016/j.rse.2013.08.029, **2014**.

Rokni, K., Ahmad, A., Selamat, A.,Hazini, S.: Water Feature Extraction and Change Detection Using Multitemporal Landsat Imagery, Remote Sens., 6, 4173-4189, doi:10.3390/rs6054173, **2014**.

Xu, H.: Modification of normalised difference water index (NDWI) to enhance open water features in remotely sensed imagery, Int. J. Remote Sens., 27, 3025-3033, doi:10.1080/01431160600589179, **2006**.

Lehner, B.,Doll, P.: Development and validation of a global database of lakes, reservoirs and wetlands, J. Hydrol. , 296, 1-22, doi:10.1016/j.jhydrol.2004.03.028, **2004**.

Jensen, J.: Remote Sensing of the Environment: An Earth Resource Perspective. Prentice Hall, Upper Saddle River, New Jersey, **2006**.

Hou, X., Feng, L., Duan, H., Chen, X., Sun, D.,Shi, K.: Fifteen-year monitoring of the turbidity dynamics in large lakes and reservoirs in the middle and lower basin of the Yangtze River, China, *Remote Sens. Environ., 190*, 107-121, doi:10.1016/j.rse.2016.12.006, **2017**.

3. The Landsat images used by GEE have a large range from 1984 to 2018. How about the uncertainties of trends / values for SDD analysis?

Firstly, in terms of the consistency and accuracy of the Landsat estimation results, some studies related to SDD have demonstrated that the estimated SDD results from Landsat 5 TM, 7 ETM + and 8 OLI images are highly comparable through comparing the Landsat-derived SDD values from the overlapping area of the images (Landsat 7 ETM + vs. Landsat 5 TM images and the Landsat 7 ETM + vs. Landsat 8 OLI images), which indicated that the use of Landsat series data with the proposed model can provide accurate long-term coverage of SDD in lakes in China (Zhang et al., 2021; Song et al., 2020; Deutsch et al., 2018; Bonansea et al., 2015; Mccullough et

al., 2013). Figure 5 demonstrates the overlap area of the Landsat TOA images in the lakes located at home and abroad and a scatterplot with a regression of reflectance values derived from the overlapping area. The model coefficients of these scatterplots reveal the good agreement between Landsat 5/7/8 images reflectance values. In our study, the big data of in-situ measured SDD spanned 15 years (2004-2018) was used to calibrate the model of SDD, where $R^2 = 0.79$, RMSE = 100.3 cm, rRMSE = 61.9%, MAE = 57.7 cm. Moreover, the average of estimated SDD in five lake regions is consistent with the distribution of in-situ measured SDD. In addition, in the manuscript of Figure 8, we also used available in-situ SDD data (2019 – 2020) collected at monitoring stations in Lake Taihu and Lake Dianchi to assess the accuracy of the model, and the result showed that our model exhibited good performance to retrieve SDD in both Lake Taihu and Lake Dianchi.

Secondly, from the perspective of uncertainties of Landsat estimation results, the effects of a few systemic errors on estimated SDD results are unavoidable. On the one hand, the SDD estimation model proposed in this study existed some errors, where the validation model showed $R^2=0.80$, RMSE = 92.7 cm, RMSE% = 57.6%, MAE= 54.9 cm. On the other hand, the atmosphere affects differently sensor bands depending on the waveband, thus affecting the relationships obtained from top-of-atmosphere reflectance, and different atmospheric correction methods cause diverse effects on the Landsat images (Bonansea et al., 2015; Lee et al., 2016). It's noted that the TOA products were produced using the Landsat Ecosystem Disturbance Adaptive Processing System (LEDAPS) software provided within GEE (Schmidt et al., 2013). These two factors may lead to the uncertainties of estimated SDD results based on Landsat long-term observation. Although systemic errors are inevitable, they do not have much impacts on the overall trend towards SDD of lakes. The revised content could be seen in the section of "7 Comparison with past studies and uncertainties" of revised-manuscript.

[Figure]

Figure 4: Overlap areas of the Landsat 7 ETM+ and Landsat 5 TM TOA images (a and c) and the Landsat 7 ETM+ and Landsat 8 OLI TOA images (b and d) in the lakes of Selin co in China and

Van Lake in Turkey, respectively. Reflectance comparison for Landsat 7 ETM+ vs. Landsat 5 TM TOA images (e and g) and the Landsat 7 ETM+ vs. Landsat 8 OLI TOA images (f and h) for these two lakes derived from the overlap.

**References:**

Zhang, Y., Zhang, Y., Shi, K., Zhou, Y.,Li, N.: Remote sensing estimation of water clarity for various lakes in China, Water Res., 192, 116844-116844, doi:10.1016/j.watres.2021.116844, **2021**.

Song, K., Liu, G., Wang, Q., Wen, Z., Lyu, L., Du, Y., Sha, L.,Fang, C.: Quantification of lake clarity in China using Landsat OLI imagery data, *Remote Sens. Environ., 243*, 111800, doi:10.1016/j.rse.2020.111800, **2020**.

Bonansea, M., Bazan, R., Ledesma, C., Rodriguez, C.,Pinotti, L.: Monitoring of regional lake water clarity using Landsat imagery, *Hydrol. Res. , 46*, 661-670, doi:10.2166/nh.2014.211, **2015**

McCullough, I. M., Loftin, C. S.,Sader, S. A.: Landsat imagery reveals declining clarity of Maine's lakes during 1995-2010, *Freshw. Sci., 32*, 741-752, doi:10.1899/12-070.1, **2013**.

Deutsch, E. S., Alameddine, I.,El-Fadel, M.: Monitoring water quality in a hypereutrophic reservoir using Landsat ETM plus and OLI sensors: how transferable are the water quality algorithms?, *Environ. Monit. Assess., 190*, doi:10.1007/s10661-018-6506-9, **2018**.

Bonansea, M., Ledesma, C., Rodriguez, C., Pinotti, L.,Antunes, M. H.: Effects of atmospheric correction of Landsat imagery on lake water clarity assessment, *Adv. Space Res., 56*, 2345-2355, doi:10.1016/j.asr.2015.09.018, **2015**.

Lee, Z., Shang, S. L., Qi, L., Yan, J.,Lin, G.: A semi-analytical scheme to estimate Secchi-disk depth from Landsat-8 measurements, *Remote Sens. Environ., 177*, 101-106, doi:10.1016/j.rse.2016.02.033, **2016**.

Schmidt, G. L., Jenkerson, C. B., Masek, J., Vermote, E.,Gao, F.: Landsat ecosystem disturbance adaptive processing system (LEDAPS) algorithm description, *U.s.geological Survey*, **2013**.

4. Causes of the regional differences and trends of water clarity, related with lake size, volume and volume changes can be added for discussion.

Response: Thank you for this suggestion, it's a good idea for analyzing the driving factors of water clarity changing trends and regional differences. But the focus of this paper is on describing the data and emphasized the data quality, validation, and utilization, so this paper didn't discuss the causes of water clarity interannual change trends. In the future, we will further investigate the influences of various potential driving factors on water clarity change trends and region differences, like natural factors (wind speed, temperature, precipitation, NDVI, water depth, water size, lake volume, DEM, slope) and anthropogenic factors (land use change, chemical fertilizer use, wastewater discharge).

**Specific comments:**

1) change the unit of ha to $km^2$

Response: Thank you for this suggestion, the all units of ha have been changed into $km^2$. Detailed revision can be seen below.

Lines 18-19: The red/blue band ratio algorithm was applied to map SDD for lakes (> 0.01 km²) based on the first SDD dataset, where $R^2 = 0.79$, RMSE = 100.3 cm, rRMSE = 61.9%, MAE = 57.7 cm.

Lines 38-39: More than 26,000 lakes (with area $> 0.01$ km²) and 78,000 reservoirs are distributed across China (Song et al., 2018), providing multiple ecosystem services (Feng et al., 2019b; Lehner and Doll, 2004; Tranvik et al., 2009; Yang and Lu, 2014).

Lines 94-95: The overall purpose of this study was to map the spatiotemporal variation of SDD in lakes ($> 0.01$ km²) across China from 1984 to 2018.

Lines 204-205: Then, combining the aforementioned image-processing methods, Eq. (1) was applied to the TOA images from 1984 to 2018 to estimate the SDD in the lakes with an area $> 0.01$ km² over China via the GEE platform.

Line 208: At last, 10,814 lakes (size $> 0.01$ km²) were used to examined for the interannual dynamics of SDD (Fig.1c).

Lines 392-394: Therefore, it becomes a challenge to compare these past results with the results of the present study due to difference in the period of interest, resolution of the satellite images and lake size ($> 0.01$ km² in our study).

**References:**

Song, K., Wen, Z., Shang, Y., Yang, H., Lyu, L., Liu, G., Fang, C., Du, J.,Zhao, Y.: Quantification of dissolved organic carbon (DOC) storage in lakes and reservoirs of mainland China, *J. Environ. Manage., 217*, 391, **2018**.

Feng, S. L., Liu, S. G., Huang, Z. H., Jing, L., Zhao, M. F., Peng, X., Yan, W. D., Wu, Y. P., Lv, Y. H., Smith, A. R., McDonald, M. A., Patil, S. D., Sarkissian, A. J., Shi, Z. H., Xia, J.,Ogbodo, U. S.: Inland water bodies in China: Features discovered in the long-term satellite data, *Proc. Natl. Acad. Sci. U. S. A., 116*, 25491-25496, doi:10.1073/pnas.1910872116, **2019b**.

Lehner, B.,Doll, P.: Development and validation of a global database of lakes, reservoirs and wetlands, *J. Hydrol. , 296*, 1-22, doi:10.1016/j.jhydrol.2004.03.028, **2004**.

Tranvik, L. J., Downing, J. A., Cotner, J. B., Loiselle, S. A., Striegl, R. G., Ballatore, T. J., Dillon, P., Finlay, K., Fortino, K., Knoll, L. B., Kortelainen, P. L., Kutser, T., Larsen, S., Laurion, I., Leech, D. M., McCallister, S. L., McKnight, D. M., Melack, J. M., Overholt, E., Porter, J. A., Prairie, Y., Renwick, W. H., Roland, F., Sherman, B. S., Schindler, D. W., Sobek, S., Tremblay, A., Vanni, M. J., Verschoor, A. M., von Wachenfeldt, E.,Weyhenmeyer, G. A.: Lakes and reservoirs as regulators of carbon cycling and climate, *Limnol. Oceanogr., 54*, 2298-2314, doi:DOI 10.4319/lo.2009.54.6_part_2.2298, **2009**.

Yang, X.,Lu, X.: Drastic change in China's lakes and reservoirs over the past decades, *Sci Rep-Uk, 4*, doi:10.1038/srep06041, **2014**.

Song, K., Liu, G., Wang, Q., Wen, Z., Lyu, L., Du, Y., Sha, L.,Fang, C.: Quantification of lake clarity in China using Landsat OLI imagery data, *Remote Sens. Environ., 243*, 111800, doi:10.1016/j.rse.2020.111800, **2020**.

2) "More than 26,000 lakes (with area >1 ha) and 78,000 reservoirs are distributed across China (Song et al., 2018)" How the 26,000 lakes (with area >1 ha) are mapped?

Response: Thank you for this careful review, it's our oversight to cite the wrong reference of Song et al. (2018a), and the proper one is Song et al. (2018b). In the right reference, it gave a detailed description of how to extract water body shoreline information in the section of "2.4.2. Small lakes or reservoirs with uncertain volume data". And in the section of "3.2.1. Lakes of each limnetic region", it gave the specific number of lakes in each lake region. The purpose of writing this sentence was to highlight the quantity of lakes and reservoirs across China, so we did not depict the method of mapping these lakes and reservoirs here.

**References:**

Song, K., Wen, Z., Xu, Y., Yang, H., Lili, L., Zhao, Y., Fang, C., Shang, Y.,Du, J.: Dissolved carbon in a large variety of lakes across five limnetic regions in China, *J. Hydrol. , 563*, 143-154, doi:10.1016/j.jhydrol.2018.05.072, **2018a**.

Song, K., Wen, Z., Shang, Y., Yang, H., Lyu, L., Liu, G., Fang, C., Du, J.,Zhao, Y.: Quantification of dissolved organic carbon (DOC) storage in lakes and reservoirs of mainland China, *J. Environ. Manage., 217*, 391, **2018b**.

Feyisa, G. L., Meilby, H., Fensholt, R.,Proud, S. R.: Automated Water Extraction Index: A new technique for surface water mapping using Landsat imagery, *Remote Sens. Environ., 140*, 23-35, doi:10.1016/j.rse.2013.08.029, **2014**.

3) "(Duan et al., 2009; Feng et al., 2019a; Kloiber et al., 2002; McCullough et al., 2012; Olmanson et al., 2011a; Pi et al., 2020; Shen et al., 2020; Song et al., 2020)." Please cite less than 5 papers at one place each time.

Response: Thank you for this suggestion, the redundant papers have been removed in line 66-67. (Duan et al., 2009; Feng et al., 2019a; McCullough et al., 2012; Olmanson et al., 2011; Shen et al., 2020)

**References:**

Duan, H., Ma, R., Zhang, Y.,Zhang, B.: Remote-sensing assessment of regional inland lake water clarity in northeast China, *Limnology, 10*, 135-141, doi:10.1007/s10201-009-0263-y, **2009**.

Feng, L., Hou, X. J.,Zheng, Y.: Monitoring and understanding the water transparency changes of fifty large lakes on the Yangtze Plain based on long-term MODIS observations, *Remote Sens. Environ., 221*, 675-686, doi:10.1016/j.rse.2018.12.007, **2019a**.

McCullough, I. M., Loftin, C. S.,Sader, S. A.: Combining lake and watershed characteristics with Landsat TM data for remote estimation of regional lake clarity, *Remote Sens. Environ., 123*, 109-115, doi:10.1016/j.rse.2012.03.006, **2012**.

Olmanson, L. G., Brezonik, P. L.,Bauer, M. E.: Evaluation of medium to low resolution satellite imagery for regional lake water quality assessments, *Water Resour. Res., 47*, doi:10.1029/2011wr011005, **2011**.

Shen, M., Duan, H., Cao, Z., Xue, K., Qi, T., Ma, J., Liu, D., Song, K., Huang, C.,Song, X.: Sentinel-3 OLCI observations of water clarity in large lakes in eastern China: Implications for SDG 6.3.2 evaluation, *Remote Sens. Environ., 247*, 111950, doi:10.1016/j.rse.2020.111950, **2020**.

4) "Regionally, lakes distribution is as follows…" Which lake data set was used?
Please state here
Response: Thank you for this suggestion, we really didn't specify the data set used here, and the source of this data set has been added in the lines 106-107: Regionally, lakes distribution sourced from Song et al. (2020) is as follows (in decreasing order): 49% in ELR, 22% in NLR, 18% in YGR, 8% in MXR and 4% in TQR (Fig. 1b).

**Reference:**

Song, K., Liu, G., Wang, Q., Wen, Z., Lyu, L., Du, Y., Sha, L.,Fang, C.: Quantification of lake clarity in China using
   Landsat OLI imagery data, *Remote Sens. Environ., 243*, 111800, doi:10.1016/j.rse.2020.111800, **2020**.

5) 1,301pairs, need a space
Response: Thank you for this suggestion, the spatial distribution of the 1,301 pairs of data used to calibrate and validate has been displayed in Fig.2a.

6) the 5% significance level to at the 5% significance level
Response: Thank you for your careful review, the phrase of "the 5% significance level" in line 228 has been changed into "at the 5% significance level".

7) Xinjiang province to Xinjiang Uygur Autonomous Region
Response: Thank you for your careful review, the "Xinjiang province" in line 271 has been changed into the "Xinjiang Uygur Autonomous Region".

Response to anonymous referee #2:

Anonymous referee #2:

The manuscript describes a water clarity product derived from Landsat images available through GEE. The manuscript is well organized and written, and the derived product would benefit the community that is interested in inland water management and the response of inland waters to climate change. Before this manuscript is published, this reviewer believes that addressing the following general comments would improve the quality of the manuscript.

1.  In the manuscript, some statements have confliction, particularly about the used Landsat images. At some places, it is stated that TOA reflectance is used, but in figure captions, surface reflectance is stated. Please clarify.

Response: Thank you for your careful review, we have carefully checked this problem in the whole manuscript, and the detailed revisions can be seen below.

1) Lines 247-248: Figure 3: Model calibration and validation for SDD estimation with Landsat TOA reflectance product acquired by different Landsat sensors. (The word of "surface" has been revised into "TOA".)

2.  Some sentences or statements are confusing, and need to be rephrased.

Response: Thank you for your careful review, the confused sentences or statements have been rephrased. Detailed revisions can be seen below.

1) Lines 77-78: "Yet, the wealth of ecological information contained in the archived Landsat images has not been fully explored." This sentence has been deleted.

2) Lines 154-155: Landsat imagery atmospheric correction is a key step for water quality inversion (Wang et al., 2009), particularly for monitoring of temporal variation at large scale. This sentence has been rephrased, that is "Landsat imagery atmospheric correction is a key step for water quality inversion (Wang et al., 2009), particularly for monitoring of temporal variation at large scale. The TOA products were produced using the Landsat Ecosystem Disturbance Adaptive Processing System (LEDAPS) software within GEE (Schmidt et al., 2013)."

3) Lines 222-224: $Y_{observed,i}$ refers to the in situ SDD measurements, $\overline{Y_{observed,i}}$ refers to the average of observed Y, and $Y_{estimated,i}$ refers to the estimated SDD from the Landsat data. (The explanation of $\overline{Y_{observed,i}}$ has been added.)

4) Lines 358-361: "In the lakes of Type III, the ELR and the TQR had the largest proportions of lake areas when the lake SDDs were between 0.5-2 m and 2-3 m, respectively, accounting for 76.80% (SDD: 0.5-1m), 46.90% (SDD: 1-2 m) and 46.65% (SDD: 2-3 m) of the total lake area in each lake region, respectively (Fig.7b-d)." This sentence has been rephrased, that is "In the lakes of Type III, the ELR had the largest proportion of lake area when SDD was 0.5-2 m, and TQR the largest when SDD was 2-3 m. The percentages of lake area when SDD was 0.5-2 m in the ELR were 76.80% (SDD: 0.5-1m) and 46.90% (SDD: 1-2 m), while that

in the TQR was 46.65% (SDD: 2-3 m)."

5) Lines 396-398: Because of the similarity of methods and images used in Zhang et al. (2021) and the present study, it provides a unique opportunity to compare in-situ measured SDD with SDD estimation obtained by Zhang et al. (2021) and in our study. This sentence has been rephrased, that is "Because of the similarity of methods and images used in Zhang et al. (2021) and the present study, it provides a unique opportunity to compare the lake SDD estimation model across China proposed by these two researches.

**References:**

Wang, M., Son, S.,Shi, W.: Evaluation of MODIS SWIR and NIR-SWIR atmospheric correction algorithms using SeaBASS data, *Remote Sens. Environ., 113*, 635-644, doi:10.1016/j.rse.2008.11.005, **2009**.

Zhang, Y., Zhang, Y., Shi, K., Zhou, Y.,Li, N.: Remote sensing estimation of water clarity for various lakes in China, *Water Res., 192*, 116844-116844, doi:10.1016/j.watres.2021.116844, **2021**.

Wang, M., Son, S.,Shi, W.: Evaluation of MODIS SWIR and NIR-SWIR atmospheric correction algorithms using SeaBASS data, *Remote Sens. Environ., 113*, 635-644, doi:10.1016/j.rse.2008.11.005, **2009**.

Schmidt, G. L., Jenkerson, C. B., Masek, J., Vermote, E.,Gao, F.: Landsat ecosystem disturbance adaptive processing system (LEDAPS) algorithm description, *U.s.geological Survey*, **2013**.

3. There are some grammar errors, and the suggestions and comments from this reviewer can be found in the annotated pdf document.

Response: Thank you for your careful review, the grammar errors have been revised. Detailed revisions can be seen below.

1) Lines 48-49: Across the country, the number of stations dedicated to the monitoring of water quality in lakes (59) and reservoirs (52) is very limited in comparison to the national inventory of lakes and reservoirs. (The word of "are" has been revised into "is".)

2) Line 50-52: Commonly expressed as Secchi disk depth (SDD) (Carlson, 1977), water clarity provides both a practical and comprehensive measure of the trophic state of aquatic ecosystems (Olmanson et al., 2008; Richardson et al., 2010). (The word of "a" between "and" and "comprehensive" has been deleted.)

3) Lines 60-61: Remote sensing has been widely used for monitoring the spatiotemporal dynamics of SDD at regional and national scales. (The word of "scale" has been revised into "scales".)

4) Lines 116-117: The percentage distribution of lakes, based on the number of lakes and lakes surface area in the five lake regions is shown in the pie charts. (The word of "are" has been revised into "is".)

5) Lines 117-118: The left one (green box) shows about all lakes extracted from

Landsat images (b), while the lower left corner one (red box) displays about lakes with SDD records more than 10 years (c). (The word of "by" has been revised into "from".)

6) Lines 172-174: For the first two datasets, SDD data derived from field surveys (2004-2018) were matched with the top of atmosphere reflectance (TOA) data collected by Landsat satellites overpassing a lake/reservoir within 7 days of field site visit. (The word of "air" has been revised into "atmosphere".)

7) Lines 177-178: For the third dataset, the cloud-free TOA images whose dates were closest to time recorded on the lake survey reports were selected to match the measured SDD, which were between May and October during the period of field survey. (The word of "date" has been revised into "dates".)

8) Line 208: At last, 10,814 lakes (size > 0.01 km²) were examined for the interannual dynamics of SDD. (The phrase of "used to examine" has been revised into "were examined for".)

9) Lines 225-226: Once the annual mean SDD maps were generated, the average of SDD for each pixel within a lake was calculated for the observation period (1984-2018). (The word of "were" has been revised into "was".)

10) Lines 244-245: Therefore, the estimation of SDD using images acquired by Landsat series of sensors provides a reliable method to examine historical trend in SDD through time series analysis. (The word of "the" has been added before "estimation".)

11) Lines 257-258: Although the number of lakes with SDD < 2 m was more numerous (80.9% of lakes), the total area of lakes with SDD between 0-0.5 m and > 4 m was the largest, accounting for 24% and 24.3% of the total area in each category, respectively (Fig. 4c). (The word of "were" has been revised into "was".)

12) Lines 266-267: The lakes in the NLR were located in the northwest and southwest of the region. In the YGR, the lakes were clustered in the southern and northeast of the region (i.e., mid-east of Sichuan province and most of Yunnan and Guangxi province). (The word of "are" has been revised into "were".)

13) Lines 276-277: (e) the proportion of lake number at different SDD levels in the five lake regions. (The word of "with" has been revised into "at".)

14) Lines 282-283: During 1984-2018, the lakes in the NLR exhibited the lowest SDD (mean: 0.60±0.09 m), followed by the ELR (mean: 1.23±0.17 m). (The

word of "the" has been added before "lakes".)

15) Lines 296-298: Among the three types of lakes — lakes with SDD showing significant increasing (Type I), decreasing (Type II) and nonsignificant (Type III) trends from 1984 to 2018, the lake SDDs in the Type I were mainly concentrated in 0.5-3 m, in the Type II were dominated by 0-2 m, and in the Type III widely distributed in 0-3 m. (The word of "were" has been added before "mainly" and the word of "the" has been added before "Type II".)

16) Line 304: The titles of the horizontal axis in Figures 5d, 5e, and 5f have been revised from "lake number" to "lake number percentage".
Lines 308-310: The proportions of lake numbers with different SDD values (0-0.5 m, 0.5-1 m, 1-2 m, 2-3 m, 3-4 m, and >4 m): (d) lakes with SDD showing significant increasing, (e) lakes with SDD showing significant decreasing, (f) lakes with SDD showing no significant trend. (The words of "showing" have been added after "SDD".)

[Figure]

Figure 5: The interannual dynamics of lake SDDs in China from 1984-2018. (a) the multi-year average SDD values of the modelled and in-situ SDDs in the five lake regions. (b) the interannual trends of mean lake SDDs in five lake regions based on the 5% significant level and slope that is the coefficient of simple linear regression. (c) the number of lakes with SDD showing statistically significant (p < 0.05) increasing (Type I), decreasing (Type II) and nonsignificant (Type III) trends. The proportions of lake numbers with different SDD values (0-0.5 m, 0.5-1 m, 1-2 m, 2-3 m, 3-4 m, and >4 m): (d) lakes with SDD showing significant increasing, (e) lakes with SDD showing significant decreasing, (f) lakes with SDD showing no significant trend.

17) Line 311: 6.2 Lake SDDs versus different lake sizes in China. (The word of "size" has been revised into "sizes".)

18) Lines 312-313: The annual mean SDD and lake area were both separated into six levels, and the proportions of lakes with different areas in each SDD category were demonstrated in Fig. 6. (The word of "the" between "in" and "Fig.6" has been deleted.)

19) Lines 319-320: Among the three types of lakes in each SDD category, there exists the similarity in the distribution of lakes with different sizes between the Type I and Type III, while that of Type II differentiated from these two types of lakes (Fig. 6). (The words of "in the" between "of" and "Type II" have been deleted.)

20) Lines 324-325: In the MXR, the number of lakes covering an area of 1-10 km$^2$ in the three types of lakes was much larger than that of other sizes among the lakes with SDDs in 0-3 m range (Fig. 6a-d). (The words of "were" and "more" have been revised into "was" and "larger", respectively. The word of "in" between "of" and "other" has been deleted.)

21) Lines 325-328: When the lake SDDs were > 3 m in this lake region, most of three types of lakes were dominated by the lakes covering an area of 0.01-1 km$^2$, apart from the lakes of Type III with SDD values > 4 m that the proportion of lakes with an area of 1-10 km$^2$ was slightly higher than that with an area of 0.01-1 km$^2$ (Fig. 6e-f). (The phrase of "in the Type III" between "4m" and "that" has been deleted.)

22) Lines 336-337: when SDDs were in the 0.5-1 m category, the number of lakes with an area between 1-10 km$^2$ and 10-50 km$^2$ was the largest, the percentages of which both were 40.00% (Fig. 6b). (The words of "were" and "most" have been revised into "was" and "largest", respectively.)

23) Lines 350-352: Spatially, the lakes in the Type I and Type III were mainly distributed in the central of the ELR, the western of the NLR, the mid-west of the TQR and the mid-east of the MXR, while those in the Type II were concentrated on the western of the NLR and eastern of the MXR. (The word of "that" has been revised into "those".)

24) Lines 361-362: In the lakes of Type II, the region that had the largest proportions of lake numbers and areas was inconsistent in each SDD category (0.5-3 m). (The word of "were" has been revised into "was".)

25) Lines 364-366: when the SDDs were in 2-3 m range, the lake number in the NLR was the largest and the total lake area in the ELR was the maximum (Fig.7d). (The word of "numbers" has been revised into "number". The phrases of "were the most" and "were the largest" have been revised into "was the largest" and "was the maximum", respectively)

26) Lines 370-371: In the Type II of lakes with SDD falling in the range 0.5-3 m, their distributions were scattered over part of the central and southeast coastal of the ELR, and southwest of the YGR (Fig.7b-d). (The phrase of "were between" has been revised into "falling in the range".)

27) Lines 374-376: Spatially, the lakes of Type I and Type III were concentrated at the junction of the ELR, YGR and MXR, the southeast coastal of the ELR, the southern of the YGR, and the western of the TQR. (The word of "the" between "of" and "Type I" has been deleted.)

28) Lines 376-377: The lakes of Type III were mainly distributed in the part of the southeast coastal of the ELR and the southern of the YGR. (The word of "were" has been added before "mainly".)

29) Lines 379-380: In the lakes of Type II, a few lakes existed in the MXR and YGR. (The phrase of "there were" has been deleted before "a few".)

30) Lines 387-388: The proportions of total lake area and lake number in each lake region were shown in the pie charts and histogram, respectively. (The word of "showed" has been revised into "shown".)

31) Line 396: Because of the similarity of methods and images used in Zhang et al. (2021) and the present study. (The words of "method" and "Zang" have been revised into "methods" and "Zhang", respectively.)

32) Lines 412-413: The dataset of water clarity of lakes developed in this study consists of one .shp file document containing the annual mean values of water clarity in each lake (size > 0.01 $km^2$) during 1990-2018, with a temporal resolution of 5-year. (The word of "time" has been revised into "temporal".)

33) Lines 426-429: In-situ water clarity data collected in lakes across China during 2004-2018 was used to calibrate and validate SDD models that incorporate top of atmosphere reflectance product and Google Earth Engine to map the

spatiotemporal dynamics of SDD over a 35-year time span (1984-2018). (The word of "air" has been revised into "atmosphere".)

**References:**

Carlson, R. E.: A Trophic State Index for Lakes, *Limnol. Oceanogr., 22*, 361-369, doi:10.2307/2834910, **1977**.

Olmanson, L. G., Bauer, M. E.,Brezonik, P. L.: A 20-year Landsat water clarity census of Minnesota's 10,000 lakes, *Remote Sens. Environ., 112*, 4086-4097, doi:10.1016/j.rse.2007.12.013, **2008**.

Richardson, T. L., Lawrenz, E., Pinckney, J. L., Guajardo, R. C., Walker, E. A., Paerl, H. W.,MacIntyre, H. L.: Spectral fluorometric characterization of phytoplankton community composition using the Algae Online Analyser (R), *Water Res., 44*, 2461-2472, doi:10.1016/j.watres.2010.01.012, **2010**.

---

## Author Response (AR3)

Manuscript Number: essd-2021-227
Manuscript Title: Water clarity annual dynamics (1984–2018) dataset across China derived from Landsat images in Google Earth Engine

Response to editor comments:
Publish subject to minor revisions
The reviewer had no further comments and recommended publication of the manuscript. However, I believe the manuscript still requires minor revision to address issues mainly in its writing. Below you may find few specific issues highlighted by the editor. I would encourage the authors to carefully check the manuscript and maybe have a professional language service to help improve the writing.
Response: Thank you for your professional suggestions, we have carefully checked the whole manuscript and found a professional language service institution to improve the writing. The detailed revision could be seen in the version of manuscript marked-up with change.

[Figure]

**EDITORIAL CERTIFICATE**

This document certifies that the manuscript listed below was edited for proper English language, grammar, punctuation, spelling, and overall style by one or more of the highly qualified native English speaking editors at EditSprings.

**Manuscript title:**

An inland water clarity dataset of China made using Landsat observation from 1984 to 2018

**Authors:**

Hui Tao, Kaishan Song, Ge Liu, Qiang Wang, Zhidan Wen, Pierre-Andre Jacinthe, Xiaofeng Xu, Jia Du, Yingxin Shang, Sijia Li, Zongming Wang, Lili Lyu, Junbin Hou, Xiang Wang, Dong Liu, Kun Shi, Baohua Zhang, Hongtao Duan

**Date Issued:**

Nov 26 2021

[Figure]

**Certificate Number:**

ES-202111201311137996

This certificate can be verified on www.editsprings.com/query.asp. This document certifies that the manuscript listed above was edited for proper English language, grammar, punctuation, spelling, and overall style by one or more of the highly qualified native English speaking editors at EditSprings. Neither the research content nor the authors' intentions were altered in any way during the editing process. Documents receiving this certification should be English-ready for publication; however, the author has the ability to accept or reject our suggestions and changes.

EditSprings provides a range of editing, translation and manuscript services for researchers and publishers around the world. Our top-quality PhD editors are all native English speakers from famous institutions cross the U.S., Britain, Canada and so on. Our editors come from nearly every research field and possess the highest qualifications to edit research manuscripts written by non-native English speakers.

1.  The title seems containing too much information. I would suggest the authors to simplify it to it to highlight the key information of the dataset. On the other hand, the dataset is about lakes, but title did not point it out.
    Response: Thank you for your valuable suggestion. We have changed the original title into a new one, i.e., An inland water clarity dataset of China made using Landsat observation from 1984 to 2018.

2. Abstract, line 22-23. "In 2018,……"the sentence is confusing. Please rephrase it.
   Response: Thank you for this careful review, we have rephrased this sentence in lines 25-27, i.e., In 2018, we found the number of lakes with SDD < 2 m accounted for the largest proportion (80.93%) of the total lakes, but the total area of lakes with SDD of <0.5 m and > 4 m were the largest, both accounting for about 24.00% of the total lakes, respectively.

3. I would suggest remove the word lake when mentioning the regions to avoid confusion. For example, changing "lakes in the Tibetan-Qinghai Plateau lake region" to "lakes in the Tibetan-Qinghai Plateau region".
   Response: Thanks for your patient review. We have removed the word of "lake" when mentioning the regions in the paper, and the detailed revision could be seen in the following:
   Lines 27-29: During 1984-2018, lakes in the Tibetan-Qinghai Plateau region (TQR) had the clearest water with an average value of 3.32±0.38 m, while that in the Northeastern region (NLR) exhibited the lowest SDD (mean: 0.60±0.09 m).

   Lines 31-35: At the five lake regions, except for the Inner Mongolia-Xinjiang region (MXR), more than half of the total lakes in every other region exhibited significant increasing trends. In the Eastern region (ELR), NLR and Yungui Plateau region (YGR), almost more than 50% of the lakes that displayed increase or decrease in SDD were mainly distributed in the area range of 0.01-1 $km^2$, whereas that in the TQR and MXR were primarily concentrated in large lakes (> 10 $km^2$).

4. I would suggest the authors replace "0-0.5 m" with "< 0.5 m" to be consistent with "> 4 m".
   Response: Thank you for this suggestion, we have replaced "0-0.5 m" with "<0.5m" in the whole manuscript (including the Figures), and the detailed revision could be seen in the following:

   Lines 25-27: In 2018, we found the number of lakes with SDD < 2 m accounted for the largest proportion (80.93%) of the total lakes, but the total area of lakes with SDD of <0.5 m and > 4 m were the largest, both accounting for about 24.00% of the total lakes, respectively.

   Lines 249-250: Based on their mean SDD, all lakes across China in 2018 were divided into six levels, i.e., <0.5 m, 0.5-1 m, 1-2 m, 2-3 m, 3-4 m, and >4 m, with 26.4%, 25.7%, 28.8%, 12.5%, 4.3%, and 2.3% of lakes in each SDD level, respectively (Fig. 4b).

   Lines 251-252: Although the number of lakes with SDD < 2 m was more numerous (80.9% of lakes), the total area of lakes with SDD of <0.5 m or > 4 m

was the largest, accounting for 24% and 24.3% of the total area in each category, respectively (Fig. 4c).

Lines 268-270: Annual mean SDD of lakes (>0.01 km²) across China in 2018. (a) Spatial distribution of lakes with SDD values. (b) Proportion of lake number with SDD values for six levels (i.e., ≤0.5 m, 0.5-1 m, 1-2 m, 2-3 m, 3-4 m, and >4 m).

[Figure]

Lines 289-291: Figure 5: The interannual dynamics of lake SDDs in China during 1984-2018. (a) Multi-year average SDD values of the modelled and in-situ SDDs in the five lake regions. (b) Interannual trends of mean lake SDDs in five lake regions based on the 5% significant level and slope representing the coefficient of simple linear regression. (c) Number of lakes with SDD showing statistically significant ($p < 0.05$) increasing (Type I), decreasing (Type II) and nonsignificant (Type III) trends. Proportions of lake numbers with different SDD values (≤0.5 m, 0.5-1 m, 1-2 m, 2-3 m, 3-4 m, and >4 m) for: (d) lakes with SDD showing significant increasing trend; (e) lakes with SDD showing significant decreasing trend; and (f) lakes with SDD showing no significant trend.

[Figure]

Lines 320-322: The lakes of Type II, located in the three lake regions, with SDD values of 0.5-1 m in the ELR, and of ≤0.5 m and 2-3 m in the NLR were dominated by the area size of 1-10 km², while the remaining lakes were mostly with the area range of 0.01-1 km² (Fig. 6a-f).

Lines 335-336: For the lakes of Type II in the TQR, the lakes with SDDs in the ≤0.5 m category were distributed in the area range of 10-50 km², followed by that of 50-100 km² (Fig. 6a).

Lines 343-345: Figure 6: Proportions of lake numbers in different areas in the six SDD categories. The six SDD categories are: (a) ≤0.5 m; (b) 0.5-1 m; (c) 1-2 m; (d) 2-3 m; (e) 3-4 m; (f) >4 m. The SDD values are the average of estimated results in each lake during 1984-2018. In the five lake regions, the lakes are further divided into three types — lakes with SDD showing significant increasing (Type I), decreasing (Type II) and nonsignificant (Type III) trends during 1984-2018.

[Figure]

Lines 349-351: In the SDD of ≤0.5 m category (Fig.7a), the NLR had the largest lake numbers and areas in the three types of lakes, accounting for 34.51% and 33.20% in Type I, 63.19% and 48.17% in Type II and 44.46% and 34.38% in Type III of the number of lakes and areas in the lake region, respectively.

Lines 386-390: Figure 7: Spatial distribution of lakes with multi-year average SDD values during 1984-2018. The SDD values were divided into six levels: (a) ≤0.5 m; (b) 0.5-1 m; (c) 1-2 m; (d) 2-3 m; (e) 3-4 m; (f) >4 m. The lakes were separated into three types of lakes—lakes with SDD showing significant increasing (Type I), decreasing (Type II) and nonsignificant (Type III) trends during 1984-2018.Proportions of total lake area and lake number in each lake region are shown in the pie charts and histogram, respectively.

[Figure]

5. Line 131. "delineating water body boundaries…" please confirm if it was to delineate the water body boundaries or simply classify water and non-water at the step?

Response: Thank you for this question, the word of "delineating" is something of a misnomer, and we have replaced it with "extract" in lines 133 and 139, i.e., We extracted the boundaries of these changing lakes using Landsat images during 1984-2018.; First, we used the MNDWI, combined with Tasseled Cap Transformation (TC) and a density slicing with multi-threshold approach, to build a decision tree for extracting water body boundaries using the ENVI software package (Rokni et al., 2014; Xu et al., 2006).

**References:**

Rokni, K., Ahmad, A., Selamat, A.,Hazini, S.: Water Feature Extraction and Change Detection Using Multitemporal Landsat Imagery, Remote Sens., 6, 4173-4189, doi:10.3390/rs6054173, **2014**.

Xu, H.: Modification of normalised difference water index (NDWI) to enhance open water features in remotely sensed imagery, Int. J. Remote Sens., 27, 3025-3033, doi:10.1080/01431160600589179, **2006**.

6. Line 140, "we mainly viewed the Landsat (5/7/8) and Google Earth images to confirm the changing region" the sentence is confusing.
   Response: Thank you for this professional question, we have described this sentence clearer than before in lines 128-130, i.e., As for the reservoirs, we mainly viewed and compared the Landsat natural color images on the website of Earthdata Search (https://search.earthdata.nasa.gov/) and historical images embedded in Google Earth to confirm the changing region, respectively.

7. Line 141, "With respect to the small lakes with an area < 1km$^2$, we assumed that their boundaries didn't change during the study period" How the small lakes boundaries was determined?
   Response: Thank you for your patient review. We may not express the meaning of this sentence clearly to make reader confused, and we have revised this sentence in lines 131-132, i.e., For the small lakes with area < 1 km$^2$ obtained from the study of Song et al. (2020) we assumed their boundaries to remain unchanged during the study period. In the beginning of this paragraph, we gave a brief introduction about lake boundaries: Following Song et al. (2020), the lake boundaries (lakes and reservoirs) with area > 0.01 km$^2$ across China were derived from Landsat 8 OLI images mainly acquired in 2016, and detailed description on boundary extraction is available in that study.

**References:**

Song, K., Liu, G., Wang, Q., Wen, Z., Lyu, L., Du, Y., Sha, L.,Fang, C.: Quantification of lake clarity in China using Landsat OLI imagery data, *Remote Sens. Environ., 243*, 111800, doi:10.1016/j.rse.2020.111800, **2020**.

8. Line 153, "We divided water bodies into lakes, reservoirs, and rivers according to their shoreline features, and also through referencing to the Global Reservoirs and Dams database (Lehner et al., 2011), Chinese Reservoirs and Dams database, and high-resolution images from Google Earth to tell rivers and reservoirs from water bodies." Maybe the authors distinguished the lakes from reservoirs, and rivers mainly by visual interpretation?
   Response: Thank you for this question, we indeed distinguished the lakes from reservoirs, and rivers mainly by visual interpretation, which took a lot of manpower and time. We have added these key words in lines 143-146, i.e., According to the shoreline features, we divided water bodies into lakes, reservoirs and rivers. By referring to the Global Reservoirs and Dams database (Lehner et al., 2011), Chinese Reservoirs and Dams database and high-resolution images from Google Earth, we distinguished rivers and reservoirs from water bodies mainly by visual interpretation.

**References:**

Lehner, B., Liermann, C. R., Revenga, C., Voeroesmarty, C., Fekete, B., Crouzet, P., Doell, P., Endejan, M., Frenken, K., Magome, J., Nilsson, C., Robertson, J. C., Roedel, R., Sindorf, N.,Wisser, D.: High-resolution mapping of the world's reservoirs and dams for sustainable river-flow management, *Front. Ecol. Environ., 9*, 494-502, doi:10.1890/100125, **2011**.

9. Line 160, "one pixel buffer inward of water boundary was removed for lakes with an area ≤ 1 km², and two pixels for lakes with an area > 1 km² in order to avoid the influence of adjacent land on water bodies." the numbers of pixels buffered in this study are not well enough explained.

   Response: Thank you for your valuable comment. We have made a detailed explanation in lines 152-156, i.e., In our study, one pixel (two pixels) buffer inward of water boundary was removed for lakes with area ≤ 1 km² (> 1 km²) in order to avoid the influence of adjacent land on water bodies that can result in mixed land-water pixels. The determination of the number of pixel buffered was referenced to the method proposed in the study of Wang et al. (2018) who made a comparison of water-leaving reflectance in the transects selected from the land-water boundaries to identify a suitable buffer distance.

**References:**

Wang, S. L., Li, J. S., Zhang, B., Spyrakos, E., Tyler, A. N., Shen, Q., Zhang, F. F., Kutser, T., Lehmann, M. K., Wu, Y. H.,Peng, D. L.: Trophic state assessment of global inland waters using a MODIS-derived Forel-Ule index, *Remote Sens. Environ., 217*, 444-460, doi:10.1016/j.rse.2018.08.026, **2018**.

10. Line 173-174. Please confirm if all of the Landsat data were processed with LEDAPS, especially the Landsat 8 OLI data. LEDAPS usually only applies to TM and ETM+.

    Response: Thank you for your professional comment. We are sorry that the improper description of TOA products was given here, and we have corrected the mistake in lines 185-188 and lines 410-411. As for the LEDAPS, it is the Landsat surface reflectance products that were atmospherically corrected from raw digital values in LEDAPS and Landsat Surface Reflectance Code (LaSRC) software (Schmidt et al., 2013; Zhang et al., 2021).

    Lines 185-188: The TOA products within GEE were produced using the equations developed by Chander et al. (2009), and the function of these equations was to convert calibrated digital numbers to absolute units of TOA reflectance. The description of Landsat TOA products could be seen on the GEE platform (https://developers.google.com/earth-engine/datasets/catalog/landsat).

    Lines 410-411: The calibrated TOA reflectance products within the GEE were produced using the equations developed by Chander et al. (2009).

**References:**

Chander, G., Markham, B. L.,Helder, D. L.: Summary of current radiometric calibration coefficients for Landsat MSS, TM, ETM+, and EO-1 ALI sensors, *Remote Sens. Environ., 113*, 893-903, doi:10.1016/j.rse.2009.01.007, **2009**.

Schmidt, G. L., Jenkerson, C. B., Masek, J., Vermote, E.,Gao, F.: Landsat ecosystem disturbance adaptive processing system (LEDAPS) algorithm description, *U.s.geological Survey*, **2013**.

Zhang, Y., Zhang, Y., Shi, K., Zhou, Y.,Li, N.: Remote sensing estimation of water clarity for various lakes in China, *Water Res., 192*, 116844-116844, doi:10.1016/j.watres.2021.116844, **2021**.

11. Line 178, missed space between 1,301 and pairs.

    Response: Thank you for pointing out this mistake, and we have corrected it in line 193, i.e., ……1,301 pairs of in-situ SDD and TOA……

12. Section 6.1 "Average and temporal trend in lakes SDD". Correct "lakes" to "lake". Also, please clarify what the average was applied to, spatially or temporally?

    Response: Thank you for this question, we have revised the title of section 6.1in line 273, i.e., 6.1 Temporal average and trend in lakes SDD.

13. Section 6.2 "Lake SDDs versus different lake sizes in China". I would suggest the authors to rephase the title. Or at least remove the word "different".

    Response: Thank you for this question, we have revised the title of section 6.2 in line 309, i.e., 6.2 Lake SDDs versus lake sizes in China.

14. Line 324, "total lake numbers" could be misleading, because the number is calculated from a group of subjects. Total number usually means another level of aggregation. For example, the total numbers of lakes in the regions is calculated as the number of lakes in all of the regions combined, instead of the number of lakes in a particular region. Please confirm and correct the expresses through the manuscript if it is needed.

    Response: Thank you for pointing out the problem of conceptual confusion, we have corrected these mistakes in the whole manuscript, and the detailed revision could be seen in the following:

    lines 348-352: The spatial distributions of lakes and their number of lakes and areas of the three types of lakes in five lake regions are presented in the Fig.7. In the SDD of <0.5 m category (Fig.7a), the NLR had the largest lake numbers and areas in the three types of lakes, accounting for 34.51% and 33.20% in Type I, 63.19% and 48.17% in Type II and 44.46% and 34.38% in Type III of the number of lakes and areas in the lake region, respectively.

    Lines 363-364: When the SDDs ranged from 1-2 m, the number of lakes and area in the ELR were the largest (Fig.7c).

    Lines 379-380, Regarding the SDD of >4 m category (Fig.7f), the TQR had the

largest lake number and area in the lakes of Type I, accounting for 39.19% of the number of lakes and 87.34% of the total lake area, respectively.

Lines 381-383: For the lakes of Type III, the YGR had the most lakes and the TQR had the largest total lake area, accounting for 40.28% of the number of lakes and 87.00% of the total lake area, respectively.

15. Line 379, "… existed some errors". Please rephase it to avoid ill expression. Also, change "validation model" to either "validation" or "model validation".
Response: Thank you for your helpful comment, we have corrected these mistakes in lines 406-309, i.e., On the one hand, the SDD estimation model proposed in this study contained some errors, where the model validation yielded these results: R2=0.80, RMSE = 92.7 cm, RMSE% = 57.6%, MAE= 54.9 cm.

16. Data avaliablity section. replace ".shp file" with "shapefile file".
Response: Thank you for pointing out this mistake, we have corrected it in line 416, i.e., The dataset of water clarity of lakes developed in this study consists of one shapefile file document containing the annual mean values of water clarity in each lake (size > 0.01 km²) during 1990-2018, with a temporal resolution of 5-year.

---

## Author Response (AR4)

Manuscript Number: essd-2021-227
Manuscript Title: Water clarity annual dynamics (1984–2018) dataset across China derived from Landsat images in Google Earth Engine

Response to editor comments:
Publish subject to minor revisions

1. I believe that the authors have made a great effort to address the comments. However, I am a little concerned with the updated title, which is the most critical piece of information of the paper. I don't think dropping the word "annual" in the title was a wise move. I would like to ask the author to reconsider it before finalizing it for publication. In my opinion, the title could be something like:
   1) An annual inland water clarity dataset of China between 1984 and 2018;
   2) A Landsat derived annual inland water clarity dataset of China between 1984 and 2018.
   Response: Thank you for your valuable suggestion. We have changed the title into the second one you provided, i.e., A Landsat derived annual inland water clarity dataset of China between 1984 and 2018.

2. Please replace "shapefile file document" with "shapefile" in the Data availability section.
   Response: Thank you for your careful review. We are sorry we made such a mistake, and we have corrected it in line 397, i.e., The dataset of water clarity of lakes developed in this study consists of one shapefile containing the annual mean values of water clarity in each lake (size > 0.01 km²) during 1990-2018, with a temporal resolution of 5-year.

3. Also, small lakes are likely to be sensitive to changes in climate or human activities, and assuming static boundaries for small lakes may impact the accuracy of SDD simulation. I would suggest the authors add a sentence or two in the discussion to clarify whether the assumption could lead to uncertainty in the dataset.
   Response: Thank you for your professional suggestion. We have considered it seriously and added two sentences in the discussion of section 7 in lines 393-397, i.e., In addition, under the influence of climate change or human activities, such as floods and droughts, urbanization, and farmland reclamation, the boundaries for some small lakes ($< 1$ km²) may vary greatly, which could cause the uncertainty of SDD estimation (Yang et al., 2021; Zhang et al., 2019). This is a limitation of the assumption for small lakes with static boundaries. In the future, further research on the relationship between the area of small lakes and the accuracy of SDD simulation would aid in addressing this limitation.

**References:**

Yang, J.,Huang, X.: The 30 m annual land cover dataset and its dynamics in China from 1990 to 2019, *Earth Syst. Sci. Data, 13*, 3907-3925, doi:10.5194/essd-13-3907-2021, **2021**.

Zhang, G., Yao, T., Chen, W., Zheng, G., Shum, C. K., Yang, K., Piao, S., Sheng, Y., Yi, S., Li, J., O'Reilly, C. M., Qi, S., Shen, S. S. P., Zhang, H.,Jia, Y.: Regional differences of lake evolution across China during 1960s-2015 and its natural and anthropogenic causes, *Remote Sens. Environ., 221*, 386-404, doi:10.1016/j.rse.2018.11.038, **2019**.